# Forgetting Whenever You Want: A Decentralized Continual Learning Framework with On-Demand Unlearning

**Xiao Zhang** [1]  **Zengzhe Chen** [1]  **Mingyi Li** [1]  **Jing Qiao** [1 2]  **Fuzhen Zhuang** [3]  **Yuan Yuan** [4]  **Dongxiao Yu** [1]

## Abstract

Decentralized class continual learning refers to a paradigm where distributed clients continuously acquire new classes while retaining previously learned information without relying on a central server. With increasing emphasis on privacy preservation, there is a growing need for on-demand unlearning, introducing two key challenges: *Historical Class Unlearning* and *Network-Wide Knowledge Entanglement*. In this work, we propose a **d**ecentralized **c**ontinual learning framework with on-demand **u**nlearning (**DCU**), which is the first attempt at achieving class continual learning and arbitrary-time class unlearning in a distributed setting. Specifically, our proposed DCU comprises three main stages: prototypes extraction, prototype-guided continual learning, and unlearning with disposable prototypes. Firstly, the prototypes extraction mechanism is designed to capture the class-specific concepts as lightweight, disposable embeddings. Then, the synthetic data guided by these prototypes can be combined with real data to achieve incremental learning through distillation. Besides, synthetic samples with noisy label are used to guide the adjustment of the model's decision boundary, effectively erasing the influence of the target class while preserving other classes' knowledge. Extensive experiments conducted on two datasets demonstrate the effectiveness of our DCU in dynamic learning and target class unlearning.

---

[1]School of Computer Science and Technology, Shandong University, Qingdao, China [2]Zhongtai Securities Institute for Financial Studies, Shandong University, Jinan, China [3]School of Artificial Intelligence, Beihang University, Beijing, China [4]School of Artificial Intelligence, Shandong University, Jinan, China. Correspondence to: Mingyi Li <limee@mail.sdu.edu.cn>, Yuan Yuan <yyuan@sdu.edu.cn>.

*Proceedings of the 43rd International Conference on Machine Learning*, Seoul, South Korea. PMLR 306, 2026. Copyright 2026 by the author(s).

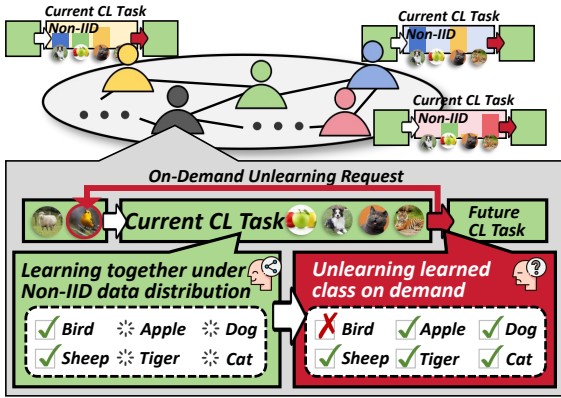

*Figure 1.* Illustration of main challenges: (1) Historical Class Unlearning - erasing learned classes (e.g., "bird") without access to original data or harming the remaining knowledge, and (2) Network-Wide Knowledge Entanglement - unlearning the learned knowledge that has globally propagated through the entire network.

## 1. Introduction

Distributed learning paradigms, including Federated Learning (McMahan et al., 2017) and Decentralized Learning (Lian et al., 2017), enable secure, privacy-preserving collaboration among multiple entities without requiring centralized data storage. Moreover, to better cope with real-world scenarios where new classes emerge dynamically, distributed class continual learning has emerged as a promising solution by learning novel class information over time while retaining prior knowledge (Hamedi et al., 2025). These approaches have demonstrated significant success in distributed data domains, including smart city applications (Lanza et al., 2023) and healthcare (Yang, 2024).

Despite its effectiveness, growing concerns over data privacy and the enforcement of strict regulations (e.g., GDPR (Voigt & Von dem Bussche, 2017) and CCPA (Bonta, 2022)) have highlighted a critical yet often overlooked requirement: ensuring the "right to be forgotten." In response to this need, the concept of Machine Unlearning has gained attention. In contrast to class continual learning, class unlearning aims to remove a specific class from the model's generalization boundary (Gu et al., 2024). Most existing class unlearning methods target either a single-node setting or a server-

orchestrated federated architecture, where all historical data remain accessible. Traditional retraining approaches (Liu et al., 2021; 2022; Tao et al., 2024) and partial-training schemes (Wang et al., 2022; Zhao et al., 2023; Ye et al., 2024) depend on full access to past samples for effective removal. Lightweight boundary-adjustment techniques such as FedAU (Gu et al., 2024) reduce this dependence but still need the raw data to be unlearned. Synthetic-data methods like QuickDrop (Dhasade et al., 2024) generate pseudo-samples, yet they rely on cross-entropy signals collected during a single and static training run, making them un-suitable for workflows with multiple sequential tasks. Even CLMUL (Chatterjee et al., 2024), which combines continual learning and unlearning in a centralized setting, presumes a data buffer and omits privacy constraints. However, as data sources become increasingly dispersed and central server cannot always be trusted, *there is an urgent need for a decentralized continual learning framework with on-demand unlearning—yet this setting remains largely unexplored*.

Decentralized continual learning inherently faces two critical challenges: catastrophic forgetting and non-IID (non-independent and identically distributed) data. Building upon these foundational issues, enabling on-demand unlearning in a decentralized continual learning setting introduces even deeper complexities. We need to address the following challenges: (1) *Historical Class Unlearning.* As the system evolves from handling a single task to a sequence of tasks, prior training data are irretrievably lost—whether due to storage constraints or strict privacy regulations. In this context, selectively erasing learned knowledge from an earlier phase (e.g., the "bird" class, as illustrated in Figure 1) becomes a non-trivial problem. Without raw data, standard retraining or fine-tuning on a superset of remaining classes is infeasible. Instead, the model must identify, isolate, and remove only the representations and parameters corresponding to the obsolete class, while preserving the performance of remaining classes (e.g., "apple", "cat", "dog", and "tiger"). (2) *Network-Wide Knowledge Entanglement.* In the decentralized environments, model updates are continuously shared and aggregated across clients, causing class-specific knowledge to propagate, permeate participant and become globally entangled throughout the network. When a request to unlearn arises, the target information is not confined to a single client but is entangled across all participants. Achieving individually forgetting knowledge for each client, without relying on a central orchestrator, is therefore profoundly challenging.

Along this line, we propose **DCU** (**d**ecentralized **c**ontinual learning framework with on-demand **u**nlearning), a novel framework that seamlessly integrates class continual learning with selective class forgetting capabilities. At its core, DCU employs pre-trained diffusion models (DMs) to generate compact class-specific prototypes that effectively cap-

ture essential class characteristics while eliminating the need for raw data storage. By combining synthetic data guided by prototypes with real data during training, DCU effectively combats non-IID data distributions and mitigates catastrophic forgetting. Besides, the disposable prototypes benefits the unlearning process. After reconstructing historical data distributions to guide class-specific unlearning, corresponding prototype can be quickly discarded, ensuring that it does not affect subsequent learning phases. Extensive experiments on two datasets demonstrate the superiority of DCU over other baselines. Our code is available at:https://github.com/DODoDoDA/DCU. Our main contributions can be summarized as follows:

- To the best of our knowledge, we are the first to consider the continual learning with on-demand unlearning in distributed settings, under the constraints of a decentralized environment and the irreversible unavailability of previous data.

- We design a diffusion-driven framework, namely DCU, to achieve both class-incremental learning and on-demand class forgetting whenever necessary. By introducing class-specific prototypes generated via pre-trained diffusion models, DCU enables these two operations in a privacy-preserving and data-free manner.

- Extensive experiments on two datasets demonstrate that DCU achieves superior performance. Without storing real data, our method consistently and effectively forgets the target knowledge while preserving the remaining knowledge across various settings.

## 2. Related Work

### 2.1. Distributed class continual learning

In class continual learning, distributed clients are required to continuously acquire new class knowledge while preserving existing class knowledge. Inspired by classic continual learning, FedWeIT (Yoon et al., 2021) introduces the concept of distributed continual learning. FedKNOW (Luopan et al., 2023) directly stores real data from past tasks for replay, which may violate data security. CFeD (Ma et al., 2022) preserves model knowledge through distillation, but this is highly dependent on the quality of the public dataset. Inspired by the data-free replay-based continual learning in DGR (Shin et al., 2017), methods like Target (Zhang et al., 2023), DDDR (Liang et al., 2024) and Lander (Tran et al., 2024) preserve previous knowledge with synthetic data.

### 2.2. Distributed class unlearning

Class unlearning in distributed settings requires clients to remove class-specific knowledge. Some methods achieve

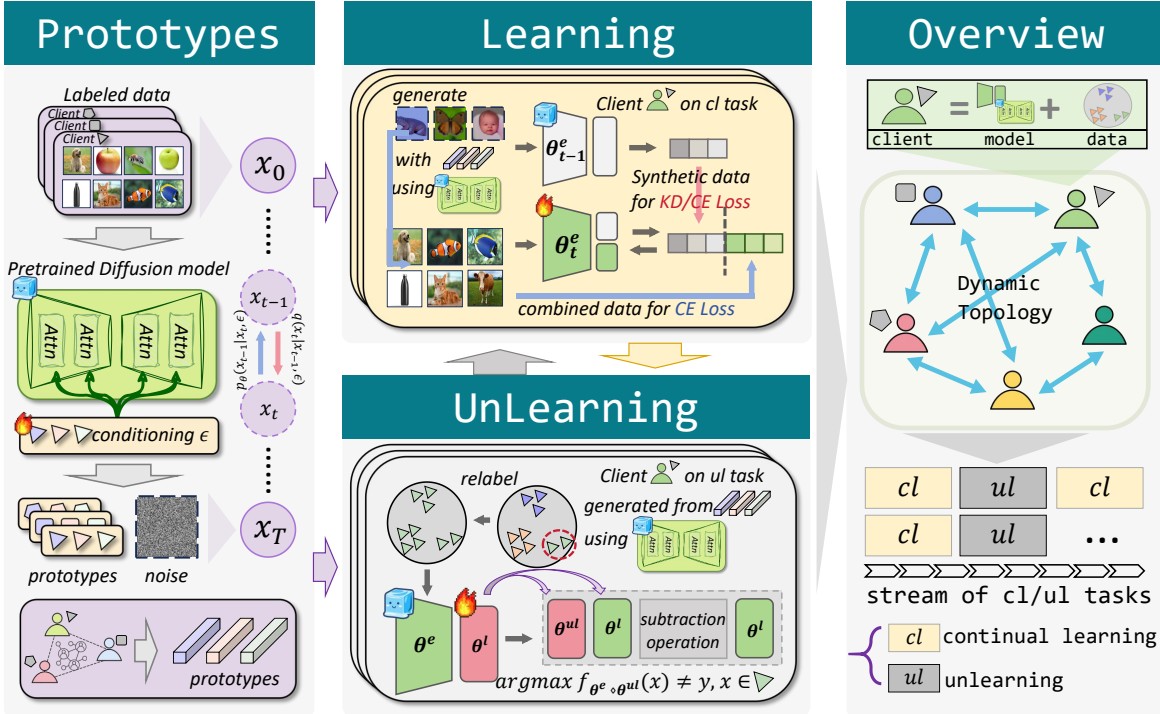

*Figure 2.* The overall framework of DCU. Before each learning task, images are processed through a frozen pre-trained diffusion model to extract prototypes for classes. In subsequent learning tasks, synthetic data that comply with the previous data distribution can be utilized to achieve knowledge transfer based on distillation. When encountering an unlearning request for the specific class, synthetic data can assist in the unlearning process, after which its corresponding prototype is discarded.

this through (partial) retraining (Liu et al., 2021; 2022; Tao et al., 2024), but at high computational cost (Liu et al., 2024). Clsprun (Wang et al., 2022) uses full-dataset sampling to guide pruning, while DoMe (Zhao et al., 2023) retrains a degenerated model on remaining data to guide unlearning, both requiring access to the entire dataset. This limits their applicability when continual and unlearning tasks intersect, as data are split into multiple subsets and cannot be accessed as an entire dataset to apply retraining. FedAF (Li et al., 2025) applies EWC-based fine-tuning with noisy labels on target data with noised label and remaining data, and FedAU (Gu et al., 2024) trains an auxiliary unlearning module on all data samples, followed by a linear operation on parameters to achieve forgetting. However, these methods still require access to learned labeled data or data distributions, restricting their application to post-task unlearning scenarios where task-specific data remain available after learning a task. QuickDrop (Dhasade et al., 2024) uses gradient ascent on synthetic data, but synthetic generation depends on original training loss, hindering removal of historical knowledge across sequential tasks. Although CLMUL (Chatterjee et al., 2024) proposes a unified continual learning and unlearning framework with a revisor-sampling buffer, it is centralized and ignores non-IID data effects and privacy constraints that prohibit storing raw data.

Diverging from these prior efforts, we introduce a novel decentralized framework for class-wise continual learning and unlearning, designed to enhance clients' generalization capability on all learned classes under non-IID settings while enabling the forgetting of specific target class knowledge at any time simultaneously.

## 3. Problem Formulation

In this work, we consider a decentralized system involving $n$ clients, denoted as $\mathcal{V} = \{1, \cdots, n\}$, which communicate over a dynamic topology represented by a doubly stochastic mixing matrix $\mathcal{W} = (W_{ij})_{n \times n}$. Here, $W_{ij} > 0$ indicates an active communication link between clients $i$ and $j$. This communication topology may change at each round, reflecting the dynamic of the network.

The above decentralized system needs to address two types of time-sensitive tasks: class continual learning and class unlearning, both of which may arrive in an unpredictable sequence. We define the set of continual learning tasks as $\mathcal{CL} = \{cl_{\mathcal{C}_t} \mid t \in \mathcal{T}\}$, where each $cl_{\mathcal{C}_t}$ corresponds to a learning task associated with class space $\mathcal{C}_t$. Notably, we consider that these class sets are strictly disjoint, i.e., $\mathcal{C}_t \cap \mathcal{C}_{t'} = \emptyset$ for all $t \neq t'$, to avoid overlap between tasks. In parallel, the system may receive class unlearn-

ing requests, captured by the set $\mathcal{UL} = \{ul_{c_f} \mid c_f \in \mathcal{C}^{(t)}\}$, where $\mathcal{C}^{(t)} = \bigcup_{i=1}^{t} \mathcal{C}_i$ denotes the cumulative class space up to time $t$. Each unlearning request $ul_{c_f}$ specifies a target class $c_f$ whose learned knowledge must be removed from the model.

For a sequence of task arrivals $\mathcal{S} = (s_1, s_2, \dots)$, each task $s_i \in \mathcal{CL} \cup \mathcal{UL}$ could either be a new learning task or a class unlearning request. (1) Each continual learning task $cl_{\mathcal{C}_t}$ comes with a dataset $\mathcal{D}_t = \{(x_i, y_i)\}$, where $x_i$ is an input image and $y_i \in \mathcal{C}_t$ is its associated class label. Data is non-IID across clients, as each client only observes a subset of the class space. (2) For each unlearning request $ul_{c_f}$, the objective is to erase the decision boundary related to class $c_f$ from the model, while maintaining the model's accuracy on the remaining classes. Specifically, the unlearned model $\Theta$ should satisfy:

$$\begin{cases} \textbf{R1: } \arg\max_b f_{\Theta_{\text{new}}}^b(x) = \arg\max_b f_{\Theta_{\text{ori}}}^b(x), & x \in \mathcal{D}_{\mathcal{C} \setminus c_f}, \\ \textbf{R2: } \arg\max_b f_{\Theta_{\text{new}}}^b(x) \neq c_f, & x \in \mathcal{D}_{c_f}. \end{cases} \quad (1)$$

Note that access to the data from previous tasks is prohibited. In other words, once a task is completed, the raw data becomes inaccessible for any future learning or unlearning operations.

Given the constraints of decentralized communication, non-IID data distribution, and the irreversible inaccessibility of past task data, the system must be capable of adapting the model in a privacy-preserving and data-free manner. This raises a fundamental challenge: how to dynamically update the model to handle continual learning and class unlearning without compromising either knowledge retention or forgetting fidelity. To this end, we formalize the objectives of this work as follows:

- **Unlearning** — to effectively eliminate knowledge of the classes specified in the unlearning requests, i.e., $\mathcal{C}_f = \{c_f\}$;

- **Retention** — to preserve the model's predictive performance on the remaining classes $\mathcal{C}^{(|\mathcal{T}|)} \setminus \mathcal{C}_f$ that were learned through prior continual learning tasks.

## 4. Methodology

As illustrated in Fig. 2, our proposed DCU includes three phases: ① *prototypes extraction* (Sec. 4.1), where pre-trained diffusion models derive class prototypes to support downstream tasks; ② *continual learning* (Sec. 4.2), which replays past knowledge via these prototypes for incremental updates; ③ *unlearning* (Sec. 4.2), where prototypes enable targeted class knowledge removal.

### 4.1. Prototypes Extraction

Considering the dynamic shifts in class distributions across tasks, replaying historical knowledge of learned classes is essential to mitigate catastrophic forgetting in continual learning. Meanwhile, for dynamic unlearning requests, enabling disposable reconstruction of historical information allows selective class forgetting without compromising subsequent continual learning processes. Building upon textual inversion based on diffusion models (DMs) (Gal et al., 2022) and following its related application in federated learning (Liang et al., 2024), we extend its application to the decentralized setting and use it to simultaneously support both learning and unlearning, introducing the approach we call Prototype Extraction.

As described in DDPM (Ho et al., 2020), the diffusion process defines a fixed Markov chain that progressively injects Gaussian noise into a clean image, eventually transforming its distribution into an isotropic Gaussian. This forward process is irrelevant of the conditioner and each step can be formulated as:

$$q(x_t \mid x_{t-1}) = \mathcal{N}\left(x_t; \sqrt{\alpha_t} x_{t-1}, (1 - \alpha_t)\mathbf{I}\right), \quad (2)$$

where $t \sim \mathcal{U}(1, T)$ denotes the timestep uniformly sampled from $[1, T]$, $x_t$ is the noisy sample at timestep $t$, and the injected noise is governed by a predefined sequence of scaling factors $\alpha_t \in (0, 1)$. In the reverse denoising process, the DMs, guided by condition, learn a denoising network $\epsilon_{\theta^d}$ to gradually reconstruct the desired image from pure Gaussian noise. In our context, since only the pre-trained DMs are used, the parameters of denoising networks are fixed. As a result, the training objective shifts from learning the denoising network to finding "prototypes" within the condition embedding space of this frozen model, which effectively capture specific class concepts. The optimization target then can be defined as:

$$p_c^* := \arg\min_p \mathbb{E}_{t,x,\epsilon \sim \mathcal{N}(\mathbf{0},\mathbf{I})} \left[ \|\epsilon - \epsilon_{\theta^d}(x_t, t, p)\|_2^2 \right], \quad (3)$$

where $x$ represents the input image, $\epsilon_{\theta^d}$ represents the frozen denoising network with parameters $\theta^d$ and the prototype $p_c$ is a trainable condition embedding for the data distribution of the $c$-th class.

In a decentralized setting, data from each class is distributed across multiple clients, and there is no trusted central server to collect and aggregate global information. After computing local prototypes, each client communicates with neighboring nodes and combines their information with its local prototypes (Lian et al., 2017) according to the communication matrix:

$$p_{c,i} = \sum_{j=1}^{n} W_{ij} p_{c,j}, \quad (4)$$

where $p_{c,i}$ denotes the prototype of $c$-th class from client $i$. Once extracted and preserved, these prototypes can be

utilized to guide the pre-trained DMs in reconstructing historical information for knowledge transfer and targeted forgetting, thereby facilitating both continual learning and on-demand unlearning. Notably, the prototypes can be safely discarded when no longer required.

## 4.2. Prototype-Guided Continual Learning

In continual learning tasks, each client faces two key challenges: acquiring knowledge from distributed new classes while mitigating catastrophic forgetting of previous knowledge. To tackle these issues, DCU introduces a Prototype-Guided Continual Learning approach that utilizes pre-trained DMs for a dual purpose: (1) it facilitates knowledge transfer from previously learned classes, and (2) it enhances learning robustness under non-IID data distributions.

Given a client classifier $\theta^{\text{full}} := \theta^e \diamond \theta^l$, composed of a feature extractor $\theta^e$ and a classification head $\theta^l$, the model undergoes adaptation when transitioning from the previous continual learning task $cl_{C_{t-1}}$ to a new task $cl_{C_t}$ that introduces $C_t$ novel classes. In this context, the classification head is expanded from $\theta^l_{t-1}$ (a $C^{(t-1)}$-way head) to $\theta^l_t$ (a $(C^{(t)}$-way head) comprising two parts: $\theta^{\hat{l}}_t$ which preserves the existing parameters of $\theta^l_{t-1}$, and $\theta^{l'}_t$ which is randomly initialized for novel classes.

To mitigate catastrophic forgetting of historical knowledge, the client employs a knowledge distillation framework that leverages replayed synthetic samples $\hat{\mathcal{D}}$, generated by conditioning a frozen pre-trained DM on class prototypes. The distillation process enforces consistency between the current model's predictions and those of the previous model, using the following Kullback-Leibler divergence objective:

$$\mathcal{L}_1 := \mathbb{E}_{\hat{x} \sim \hat{\mathcal{D}}}[KL(f_{\theta^e_t \diamond \theta^{\hat{l}}_t}(\hat{x}), f_{\theta^{full}_{t-1}}(\hat{x}))], \qquad (5)$$

where $\hat{x}$ denotes the synthetic image sample from $\hat{\mathcal{D}}$, $f_{\theta^e_t \diamond \theta^{\hat{l}}_t}$ is the student model composed of the feature extractor $\theta^e_t$ and classification head $\theta^{\hat{l}}_t$ for learned knowledge, and $f_{\theta^{full}_{t-1}}$ represents the previous-stage full model. The function $KL(\cdot, \cdot)$ computes the Kullback-Leibler divergence between the student's and teacher's output distributions. Since we employ conditional generation, all synthesized samples come with corresponding labels, enabling the use of the weighted cross-entropy loss, defined as:

$$\mathcal{L}_2 := \mathbb{E}_{(\hat{x},\hat{y}) \sim \hat{\mathcal{D}}} \left[ \sigma \left( f_{\theta^{full}_{t-1}}(\hat{x}) \right)_{\hat{y}} \cdot \ell(f_{\theta^e_t \diamond \theta^{\hat{l}}_t}(\hat{x}), \hat{y}) \right], \quad (6)$$

where $\sigma(f_{\theta^{full}_{t-1}}(\hat{x}))_{\hat{y}}$ denotes the predictive probability of the teacher model on $\hat{y}$, which serves as an adaptive weight to modulate the contribution of each sample to the loss. Here, $(\hat{x}, \hat{y})$ denotes the labeled synthetic samples from $\hat{\mathcal{D}}$ and $\ell(\cdot, \cdot)$ is the standard cross-entropy loss function.

To mitigate the impacts posed by non-IID data distributions and enhance knowledge transfer across clients with data subset of different classes, each client generates synthetic samples $\mathcal{D}'$ specifically tailored for the current learning task by conditioning the frozen pre-trained DM on prototypes. These synthetic samples are combined with the local data distribution $\mathcal{D}_t$ to minimize the follow cross-entropy loss function:

$$\mathcal{L}_3 := \mathbb{E}_{(x',y') \sim \mathcal{D}_t \cup \mathcal{D}'}[\ell(f_{\theta^e_t \diamond \theta^{l'}_t}(x'), y')], \qquad (7)$$

where $(x', y')$ denotes the training samples including features and hard labels from combined data $\mathcal{D}_t \cup \mathcal{D}'$ and $f_{\theta^e_t \diamond \theta^{l'}_t}$ represents the combination of the feature extractor $\theta^e_t$ and classification head $\theta^{l'}_t$ for new knowledge.

By combining these loss terms, DCU seeks to optimize the model $\theta^e \diamond \theta^l$, achieving balance between knowledge retention and acquisition. This is formally expressed as:

$$\mathcal{L}_{cl} := \mathcal{L}_1 + \mathcal{L}_2 + \beta_{cl}\mathcal{L}_3, \qquad (8)$$

where $\beta_{cl}$ denotes the hyperparameter associated with knowledge distillation. After updating local classifier, we employ the D-PSGD (Lian et al., 2017) to communicate clients in the dynamic topology. For T consecutive tasks, the whole optimization problem for distributed continual learning is:

$$\min f_{1:T}(\theta_T) = \frac{1}{n} \sum_{t=1}^{T} \sum_{i=1}^{n} f_{t,i}(\theta_T). \qquad (9)$$

Under the condition that the learning rate is set as $\sqrt{n/Q_l}$ and the total number of local iterations satisfies $Q_l \geq \frac{36L^2n^2}{(1-\sqrt{\rho})^2}$, our algorithm achieves the following convergence guarantee:

$$\frac{1}{Q_l} \sum_{q_l=0}^{Q_l-1} \mathbb{E}\|\nabla f_{1:T}(\frac{\Theta_{T,q_l}\mathbf{1}_n}{n})\|^2 \leq \mathcal{O}(\frac{1}{\sqrt{nQ_l}} + \frac{1}{Q_l}). \quad (10)$$

A complete proof and detailed assumptions are provided in the Appendix.

## 4.3. Unlearning with Disposable Prototypes

Compared to traditional UL methods, which typically operate in settings where unlearning is performed after a single learning phase and direct access to target data is available for removal, our setting introduces unique challenges. In our scenario, sequential learning tasks are involved, and historical data from previous tasks become inaccessible due to privacy constraints.

To address these challenges, we follow the approach of FedAU (Gu et al., 2024), which employs lightweight linear

*Table 1.* Accuracy comparison on *RGB* under continual learning with unlearning requests: ① $cl_{0:9}$(learn classes 0-9) → ② $cl_{10:19}$(learn classes 10-19) → ③ $ul_0$ (forget class 0) → ④ $cl_{20:29}$ (learn classes 20-29) → ⑤ $ul_{20}$ (forget class 20). '-' means this method doesn't apply for the experiment setting. $BS$ means the buffer size for CLMUL.

| Algorithm | $Dir(\gamma = 0.5)$ | | | | | | $Dir(\gamma = 0.1)$ | | | | | |
|---|---|---|---|---|---|---|---|---|---|---|---|---|
| | $cl_{0:9}$ (r↑%) | $cl_{10:19}$ (r↑%) | $ul_0$ (r↑/u↓%) | $cl_{20:29}$ (r↑%) | $ul_{20}$ (r↑/u↓%) | $F$ (↑) | $cl_{0:9}$ (r↑%) | $cl_{10:19}$ (r↑%) | $ul_0$ (r↑/u↓%) | $cl_{20:29}$ (r↑%) | $ul_{20}$ (r↑/u↓%) | $F$ (↑) |
| *continual learning* | | | | | | | | | | | | |
| FT | 70.40 | 35.29 | - | 24.42 | - | -0.03 | 44.94 | 22.45 | - | 17.96 | - | -0.02 |
| EWC | 75.90 | 41.10 | - | 32.97 | - | 0.05 | 44.30 | 25.70 | - | 23.10 | - | 0.20 |
| LwF | 76.50 | 53.35 | - | 44.23 | - | 0.12 | 40.70 | 30.55 | - | 26.17 | - | 0.05 |
| Target | 76.50 | 55.95 | - | 46.97 | - | 0.13 | 40.70 | 33.80 | - | 29.57 | - | 0.13 |
| Lander | 77.60 | 61.65 | - | 54.53 | - | -0.06 | 52.90 | 38.30 | - | 36.56 | - | 0.08 |
| *only unlearning recent knowledge after learning* | | | | | | | | | | | | |
| FedAF | 75.90 | 41.10 | 41.10/2.00 | 32.97 | 14.55/15.00 | 0.14 | 44.30 | 25.70 | 25.70/36.00 | 23.10 | 17.93/33.50 | 0.28 |
| EWC / AU | 75.90 | 41.10 | 41.10/2.00 | 32.97 | 28.66/0.00 | 0.43 | 44.30 | 25.70 | 25.70/36.00 | 23.10 | 19.59/2.50 | 0.62 |
| LWF / AU | 76.50 | 53.35 | 53.35/23.00 | 44.23 | 38.31/1.00 | 0.51 | 40.70 | 30.55 | 30.55/50.00 | 26.17 | 22.28/13.00 | 0.48 |
| *unlearning with continual learning using real data* | | | | | | | | | | | | |
| EWC / QDsga | 75.90 | 41.10 | 24.68/0.00 | 36.31 | 20.75/0.00 | 0.35 | 44.30 | 25.70 | 13.84/0.00 | 21.72 | 16.32/0.00 | 0.55 |
| LWF / QDsga | 76.50 | 53.35 | 30.16/0.00 | 28.69 | 5.29/0.00 | 0.22 | 40.70 | 30.55 | 16.00/0.00 | 14.59 | 4.93/0.00 | 0.33 |
| CLMUL($BS_{200}$) | 76.50 | 59.20 | 46.79/39.00 | 53.65 | 33.11/0.00 | 0.49 | 40.70 | 31.30 | 18.47/0.00 | 41.28 | 32.04/0.00 | 0.54 |
| *unlearning with continual learning using synthetic data* | | | | | | | | | | | | |
| Target / AU | 76.50 | 55.95 | 54.16/1.00 | 43.21 | 39.68/1.00 | 0.56 | 40.70 | 33.80 | 28.16/0.00 | 26.00 | 20.68/0.00 | 0.58 |
| Lander / AU | 77.60 | 61.65 | 60.95/0.00 | 52.17 | 50.96/0.00 | 0.67 | 52.90 | 38.30 | 39.58/0.00 | 35.62 | 34.11/0.00 | 0.47 |
| Ours | 75.14 | 59.44 | 58.77/0.00 | 51.48 | 50.67/0.00 | 0.66 | 65.14 | 51.45 | 47.60/0.00 | 42.94 | 40.04/0.00 | 0.59 |

operations for efficient forgetting on the decision boundary. This is coordinated by the use of our extracted prototypes, which encode the historical data distribution, to achieve both computationally efficient and flexible forgetting during continual learning.

Since the prototypes compactly capture the key concepts of historical classes, we approximate the forgetting effect on class-specific data through the unlearning of synthetic data. Specially, guided by the available prototypes, each client first utilizes the pre-trained DM to generate synthetic data $\hat{\mathcal{D}}_f$ that contains all the cumulatively learned knowledge. Then, each client performs local fine-tuning by freezing the feature extractor $\theta^e$ and optimizing the unlearning head $\theta_{ul}^l$, initialized from the original classification head $\theta_{ori}^l$. The optimization objective can be formally expressed as:

$$\mathcal{L}_{ul} := \mathbb{E}_{\hat{x} \sim \hat{\mathcal{D}}_f}[\ell(f_{\theta^e \diamond \theta_{ul}^l}(\hat{x}), c_f)], \quad (11)$$

where all synthetic samples $\hat{x} \sim \hat{\mathcal{D}}_f$ are assigned the target unlearning class label $c_f$, thereby distorting the decision boundary in the unlearning module $\theta_{ul}^l$. After that, each client performs an efficient removal of the targeted information through a subtraction operation between unlearning module $\theta_{ul}^l$ and original $\theta_{ori}^l$:

$$\theta_{new}^l := \theta_{ori}^l - \beta_{ul}\theta_{ul}^l, \quad (12)$$

where $\theta_{new}^l$ denotes the updated classification head after unlearning, and $\beta_{ul}$ denotes hyperparameter. The prototypes

are designed to be disposable, and the target prototypes corresponding to $c_f$ must be removed after this unlearning phase to prevent interference with subsequent requests. We provide the corresponding theoretical analysis in the appendix to prove that unlearning satisfies Eq. 1.

## 5. Experiments

This section presents a comprehensive evaluation of our proposed DCU with various methods. We first detail the experimental setup and comparison baselines, followed by performance analysis.

### 5.1. Experimental setup

**Dataset & Client setting** We employ two distinct datasets: 1) *Grayscale*, a unified grayscale benchmark combining the first 10 classes from each of *MNIST* (LeCun et al., 1998), *EMNIST* (Cohen et al., 2017), and *Fashion-MNIST* (Xiao et al., 2017) and 2) *RGB*, a color image dataset containing the first 30 classes of *CIFAR-100* (Krizhevsky et al., 2009). Regarding the class-incremental learning scenario, all classes from each dataset are equally partitioned into multiple tasks, here we set this to 3 tasks. To simulate heterogeneous data distributions among all clients in single task, the degree of non-IIDness is described by the Dirichlet distribution $Dir(\gamma)$ (Hsu et al., 2019). In our experiments, we set $\gamma = 0.5$ and 0.1, respectively. For the *Grayscale*

*Table 2.* Accuracy comparison on *Grayscale* under continual learning with unlearning requests: ① $cl_{0:9}$(learn classes 0-9) → ② $ul_2$ (forget class 2) → ③ $cl_{10:19}$(learn classes 10-19) → ④ $ul_0$ (forget class 0) → ⑤ $cl_{20:29}$ (learn 20-29) → ⑥ $ul_{29}$ (forget class 29). '-' means this method doesn't apply for the experiment setting. $BS$ means the buffer size for CLMUL.

| Algorithm | $Dir(\gamma = 0.5)$ | | | | | | | $Dir(\gamma = 0.1)$ | | | | | | |
|---|---|---|---|---|---|---|---|---|---|---|---|---|---|---|
| | $cl_{0:9}$ (r↑%) | $ul_2$ (r↑/u↓%) | $cl_{10:19}$ (r↑%) | $ul_0$ (r↑/u↓%) | $cl_{20:29}$ (r↑%) | $ul_{29}$ (r↑/u↓%) | $F$ (↑) | $cl_{0:9}$ (r↑%) | $ul_2$ (r↑/u↓%) | $cl_{10:19}$ (r↑%) | $ul_0$ (r↑/u↓%) | $cl_{20:29}$ (r↑%) | $ul_{29}$ (r↑/u↓%) | $F$ (↑) |
| *continual learning* | | | | | | | | | | | | | | |
| FT | 92.81 | - | 47.08 | - | 26.88 | - | 0.02 | 75.42 | - | 36.18 | - | 22.66 | - | 0.11 |
| EWC | 96.38 | - | 83.82 | - | 61.04 | - | -0.11 | 89.34 | - | 75.61 | - | 53.69 | - | 0.05 |
| LwF | 96.40 | - | 88.71 | - | 76.19 | - | -0.05 | 89.04 | - | 73.35 | - | 60.95 | - | -0.04 |
| Target | 96.40 | - | 88.62 | - | 79.37 | - | -0.04 | 89.04 | - | 76.41 | - | 63.11 | - | -0.03 |
| Lander | - | - | - | - | - | - | - | - | - | - | - | - | - | - |
| *only unlearning recent knowledge after learning* | | | | | | | | | | | | | | |
| FedAF | 96.38 | 71.84/0.00 | 76.72 | 76.72/44.00 | 48.00 | 45.83/28.93 | 0.22 | 89.34 | 84.78/0.00 | 77.24 | 77.24/42.60 | 50.03 | 54.14/36.40 | 0.32 |
| EWC / AU | 96.38 | 96.11/0.00 | 84.36 | 84.36/49.40 | 60.51 | 44.57/17.34 | 0.36 | 89.34 | 87.02/63.60 | 74.59 | 74.59/50.30 | 52.61 | 43.95/6.13 | 0.53 |
| LWF / AU | 96.40 | 96.44/0.00 | 88.65 | 88.65/49.00 | 75.29 | 59.61/28.06 | 0.41 | 89.04 | 87.22/63.80 | 76.46 | 76.46/74.10 | 60.79 | 51.44/30.20 | 0.36 |
| *unlearning with continual learning using real data* | | | | | | | | | | | | | | |
| EWC / QDsga | 96.38 | 92.47/0.00 | 86.29 | 78.96/0.00 | 59.56 | 49.07/0.00 | 0.57 | 89.34 | 75.67/0.00 | 73.51 | 61.34/0.00 | 52.34 | 24.36/0.00 | 0.37 |
| LWF / QDsga | 96.40 | 94.24/0.00 | 86.15 | 78.36/0.00 | 71.19 | 60.67/0.00 | 0.68 | 89.04 | 76.67/0.80 | 67.67 | 44.90/0.00 | 42.17 | 3.71/0.00 | 0.18 |
| CLMUL($BS_{200}$) | 96.40 | 96.11/9.40 | 95.46 | 95.10/15.40 | 90.96 | 90.04/14.00 | 0.80 | 89.04 | 88.07/59.60 | 89.53 | 89.07/60.00 | 84.11 | 83.96/46.93 | 0.45 |
| *unlearning with continual learning using synthetic data* | | | | | | | | | | | | | | |
| Target / AU | 96.40 | 96.71/0.00 | 88.36 | 89.12/0.00 | 75.30 | 74.33/0.00 | 0.82 | 89.04 | 87.49/12.60 | 76.43 | 75.50/4.10 | 59.04 | 57.90/1.46 | 0.71 |
| Lander / AU | - | - | - | - | - | - | - | - | - | - | - | - | - | - |
| Ours | 96.88 | 96.58/0.00 | 91.83 | 90.99/0.00 | 87.90 | 86.50/0.00 | 0.92 | 94.28 | 92.58/0.02 | 89.05 | 86.86/0.00 | 77.53 | 70.72/0.00 | 0.73 |

(a) *Grayscale* Dir(γ=0.5)  (b) *Grayscale* Dir(γ=0.1)  (c) *RGB* Dir(γ=0.5)  (d) *RGB* Dir(γ=0.1)

*Figure 3.* Comprehensive comparison: accuracy on remaining learned classes vs. accuracy on unlearned classes.

setting, we utiliz the test sets of MNIST, EMNIST, and Fashion-MNIST as the evaluation datasets. For the *RGB* setting, we employ the corresponding CIFAR-100 test dataset. We set client number $n = 10$ for *Grayscale*, and 5 for *RGB*, following FCL practice (Zhang et al., 2023; Qi et al., 2023; Liang et al., 2024). Configuration details are provided in the Appendix.B.

**Baselines** Firstly, we select several learning frameworks, including original *FT* (Lian et al., 2017) and federated variants of classic continual learning methods, such as *EWC* (Lee et al., 2017) and *LwF* (Li & Hoiem, 2017). We include the latest federated continual learning methods, *Target* (Zhang et al., 2023) and *Lander* (Tran et al., 2024). Then, several unlearning methods are chosen: *FedAF* (Li et al., 2025), *FedAU* (Gu et al., 2024) and SGA from *QuickDrop* (Dhasade et al., 2024). These distributed learning and unlearning approaches are combined to form our baselines. *CLMUL* (Chatterjee et al., 2024), a centralized framework for cl-ul with buffer, is also modified in a distributed manner, serving as our baseline. Detailed description provided in the

Appendix.B.1.

**Evaluation metrics** Following *CLMUL*, we evaluate the performance on remaining learned classes *r%* and unlearned classes *u%*. We also introduce the Forgetting Measure (Chaudhry et al., 2018). This metric computes the discrepancy between the maximum knowledge acquired during training and the model's current retention level, where values approaching 0 indicate optimal knowledge preservation while values near 1 signify complete forgetting. To comprehensively evaluate the model's ability to simultaneously maintain learned knowledge while selectively forgetting target classes, we define $F$ as the difference between the forgetting measures computed on unlearned classes and remaining classes.

### 5.2. Performance evaluation

The detailed results for each stage of learning and unlearning are presented in Tab. 1 and Tab. 2. Based on these results, we make the following observations.

*Table 3.* Ablation studies on *RGB* dataset under $Dir(\gamma = 0.5)$.

| $\mathcal{D}'$ | $\hat{\mathcal{D}}$ | $\hat{\mathcal{D}}_f$ | $cl_{0:9}$ ($r\uparrow\%$) | $cl_{10:19}$ ($r\uparrow\%$) | $ul_0$ ($r\uparrow/u\downarrow\%$) | $cl_{20:29}$ ($r\uparrow\%$) | $ul_{20}$ ($r\uparrow/u\downarrow\%$) |
|---|---|---|---|---|---|---|---|
| | ✓ | ✓ | 69.60 | 52.94 | 49.30(0.00) | 40.77 | 36.05(0.00) |
| ✓ | | ✓ | 75.14 | 42.51 | 42.85(0.00) | 37.52 | 33.03(0.00) |
| ✓ | ✓ | | 75.14 | 59.44 | 59.44(83.80) | 52.69 | 46.44(36.00) |
| ✓ | | | 75.14 | 42.51 | 34.76(16.00) | 34.76 | 28.10(4.80) |
| | ✓ | | 69.60 | 52.94 | 52.94(71.80) | 45.10 | 39.50(24.40) |
| | | ✓ | 69.60 | 37.80 | 35.79(0.00) | 30.15 | 25.84(0.00) |
| ✓ | ✓ | ✓ | 75.14 | 59.44 | 58.77(0.00) | 51.48 | 50.67(0.00) |

DCU achieves strong unlearning performance with high knowledge retention. It achieves high accuracy on the remaining classes (86.50%/70.72% on *Grayscale*, 50.67%/40.04% on *RGB*) and near-zero accuracy on the unlearned classes. As shown in Fig. 3, DCU consistently lies in the lower-right region, reflecting its balance between retaining useful knowledge and effectively erasing target classes. This is further confirmed by its large forgetting measure gaps between the remaining knowledge and the unlearned knowledge (0.92/0.73 on *Grayscale*, 0.66/0.59 on *RGB*), highlighting the dual strength in knowledge preservation and targeted forgetting. Note that although CLMUL achieves slightly higher accuracy on remaining classes for the single-channel image dataset *Grayscale* by employing an experience-replay method that stores real data in a buffer, this approach fundamentally violates data privacy constraints in our context. Moreover, on the *RGB* dataset under $Dir(\gamma = 0.5)$, our method achieves slightly lower performance compared to *Lander/AU*'s 50.96%. However, this generator-based method is more sensitive to the degree of data non-IIDness and exhibits less stable performance. In contrast, DCU demonstrates consistently robust and stable performance under different levels of non-IIDness.

Standard continual learning methods lack unlearning capability. Approaches such as *EWC*, *LwF*, *Target* and *Lander* perform well on retaining knowledge of remaining classes but fundamentally lack unlearning capabilities, resulting in their poorest Forgetting Measure. This is because these methods are not designed for unlearning and thus fail to effectively remove the targeted knowledge.

Most existing unlearning methods are limited in historical class removal. Techniques like *FedAF*, *EWC/AU* and *LwF/AU* maintain identical accuracy on remaining classes between the $cl_{10:19}$ and $ul_0$ stage. These approaches are designed to remove only the most recently learned knowledge prior to the unlearning request—knowledge for which access to the target data is still available. However, they fail when tasked with unlearning historical knowledge acquired in earlier learning stages. Consequently, they are ineffective at removing historical knowledge learned in earlier stages.

*QDsga* performs unlearning by solely applying gradient ascent on the target data. While it always achieves decent immediate forgetting performance (0.00%) after the first

learning task $cl_{0:9}$, the damage to the performance of the remaining classes grows increasingly severe as subsequent learning tasks arrive. *CLMUL* utilizes a reservoir sampling-based buffer to store real data for both experience replay and unlearning operations. However, this mechanism was originally designed for centralized scenarios. In decentralized settings, the non-IID nature of clients' local data distributions affects the buffer storage process on each client, making it difficult to retain historical data samples that fully represent the overall data distribution. Moreover, reservoir sampling methods progressively reduce the number of historical data samples from earlier stages, further degrading the representativeness of the stored data with respect to the global data distribution. This dual effect substantially compromises the model's ability to recover performance after unlearning, leading the significant performance degradation (33.11%/0.00% and 32.04%/0.00%) when handling more challenging dataset *RGB*. Regarding *Target/AU* and *Lander/AU*, these methods train a generator to synthesize data within a federated learning framework, enabling them to reconstruct historical information for unlearning. Specifically, Target/AU attains accuracies of 74.33% 57.90% for *Grayscale* and 39.68% 20.68% for *RGB* on remaining classes. *Lander/AU* cannot be applied to the *Grayscale* dataset due to its requirement for dataset-specific pre-trained label embeddings. However, it achieves excellent performance on *RGB*, with accuracies of 50.96% and 34.11% on the remaining classes, and 0.00% on the unlearned classes.

### 5.3. Ablation experiment

In this section, we investigate the effectiveness of different parts in DCU, including $\mathcal{D}'$, $\hat{\mathcal{D}}$ and $\hat{\mathcal{D}}_f$. Experiment is conducted on the *RGB*. The results of the ablation study experiment are shown in Table 4. We can obtain the following observation.

The absence of $\mathcal{D}'$ leads to a noticeable drop in performance when learning new knowledge, which further affects subsequent unlearning. Specifically, the accuracy on a learning task $cl_{0:9}$ decreases by approximately 5%, indicating that it impairs the model's ability to learn knowledge within a single learning task. This suggests that $\mathcal{D}'$ facilitates knowledge transfer across clients and helps the model combat non-IID data distribution. Similarly, removing $\hat{\mathcal{D}}$ results in a decline in performance when transitioning from one learning task to the next. The comparison of the performance in $cl_{10:19}$ indicates that this component effectively helps the model retain knowledge of old classes while learning new class knowledge. Furthermore, the removal of $\hat{\mathcal{D}}_f$ impairs the model's unlearning capability, particularly for forgetting historical data. The results demonstrate that this component plays a critical role in refining data distribution to help forget certain targeted data that can no longer be directly accessed.

# 6. Conclusion

In this work, we focused on the challenge of selective class forgetting during collaborative sequential learning without storing real data. We proposed DCU, a distillation based decentralized learning framework for simultaneous continual learning and on-demand unlearning. Firstly, class-specific prototypes were extracted via pre-trained diffusion models to capture the concepts of each class. These prototypes guided the generation of synthetic data, which was combined with real data for continual learning, helping to mitigate catastrophic forgetting and non-IID data challenges. Additionally, these prototypes enabled historical information reconstruction for on-demand class unlearning, after which the corresponding prototype could be discarded. The experiments conducted on two datasets demonstrated that the proposed DUC outperformed existing methods.

# Acknowledgements

This work was supported in part by Major Basic Research Program of Shandong Provincial Natural Science Foundation under Grant ZR2025ZD18, in part by National Natural Science Foundation of China (NSFC) under Grant 62302247, in part by Joint Key Funds of National Natural Science Foundation of China under Grant U23A20302, in part by the Fundamental Research Funds for the Central Universities, in part by the the Youth Innovation Technology Support Program of Higher Education Institutions in Shandong Province (Grant No. 2024KJH169).

# Impact Statement

This paper presents work whose goal is to advance the field of Machine Learning. There are many potential societal consequences of our work, none which we feel must be specifically highlighted here.

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

---

**Algorithm 1** Decentralized Continual Learning with on-demand Unlearning (DCU) Framework

---

**Require:** Initial classifiers $\{\theta^{\text{full}}\}$, task sequence $\mathcal{S}$, weight matrix $W$, and numbers of iterations $Q_p, Q_l, Q_f$

 1: **Algorithm** DCU:
 2: **for** each $s_i$ in $\mathcal{S}$ **do**
 3:    **if** $s_i = cl_{C_t} \in \mathcal{CL}$ **then**
 4:       Execute PROTOTYPEEXTRACTION on all clients
 5:       Each client generates synthetic data $\hat{\mathcal{D}}_t$ and $\mathcal{D}'_t$ guided by prototypes
 6:       Execute LEARN on all clients
 7:    **end if**
 8:    **if** $s_i = ul_{c_f} \in \mathcal{UL}$ **then**
 9:       Each client generates synthetic data $\hat{\mathcal{D}}_f$ guided by prototypes
10:       Execute UNLEARN on all clients
11:    **end if**
12:    Each client saves the copy of its $\theta^{\text{full}}$ as the teacher model
13: **end for**
14: **Output:** $\frac{1}{n} \sum_{i=1}^{N} \theta_i^{\text{full}}$

---

*Table 4.* Frequently used notations

| Notation | Description |
|----------|-------------|
| $\mathcal{S}$ | Task sequence |
| $cl_{C_t} \in \mathcal{CL}$ | Learning task with class space $C_t$ |
| $ul_{c_f} \in \mathcal{UL}$ | Unlearning task for class $c_f$ |
| $\hat{\mathcal{D}}_t$ | Synthetic data complying with the historical knowledge for learning |
| $\mathcal{D}'_t$ | Synthetic data to mitigate the influence of non-IID data distribution |
| $\mathcal{D}_f$ | Synthetic data complying with the previous data distribution for unlearning |
| $\theta^{full}$ | Full model parameters |
| $\theta^e$ | Feature extractor |
| $\theta^l$ | Classification head |
| $\theta^{ul}$ | Unlearning module |
| $\mathcal{W}$ | Weight matrix |

# A. Algorithm

To better demonstrate the proposed DCU, we provide the detailed algorithm (shown in Algo. 1 and Algo. 2) and a notation table (shown in Tab. 4).

# B. Experiment Details

### B.1. Baselines

- **EWC** (Elastic Weight Consolidation): A classic regularization-based continual learning approach that preserves important parameters for previous tasks by constraining their updates based on Fisher information.

- **LwF** (Learning without Forgetting): A classic distillation-based method that maintains performance on old tasks by using the original network's outputs as soft targets when learning new tasks.

- **Target**: This represents a state-of-the-art approach in federated continual learning. A central data generator is trained on the server using local models, and each client mitigates catastrophic forgetting by integrating synthetic unlabeled data with real data during training.

---

**Algorithm 2** Functional Modules of DCU: Prototype Extraction, Learning, and Unlearning

---

1: **Function** PROTOTYPEEXTRACTION:
2: **for** $q_p = 0$ to $Q_p - 1$ **do**
3:    **if** $q_p = 0$ **then**
4:       Randomly initialize prototypes $P_{C_t,i} = \{p_{c,i}\}$ for new class space $C_t$
5:    **end if**
6:    Update the local prototypes $P_{C_t,i}$                                       ▷ Eq.3
7:    Compute the weighted average with received neighborhood prototypes $P_{C_t,i} = \sum_{j=1}^{n} W_{ij} P_{C_t,j}$
8: **end for**
9:
10: **Function** LEARN:
11: **for** $q_l = 0$ to $Q_l - 1$ **do**
12:    **if** $q_l = 0$ **then**
13:       Expand the classification head from $\theta_{t-1,i}^{l}$ to $\theta_{t,i}^{l}$ for novel classes
14:    **end if**
15:    Randomly sample $\xi$, $\hat{\xi}$, and $\xi'$ from local data, $\hat{\mathcal{D}}_t$, and $\mathcal{D}'_t$, respectively
16:    Compute the local stochastic gradient $\nabla\mathcal{L}_{cl}(\theta_{t,i,q_l}^{\text{full}}; \xi, \xi', \hat{\xi})$                 ▷ Eq. 8
17:    Compute the weighted average with received neighborhood classifier $\theta_{t,i,q_{l+\frac{1}{2}}}^{full} = \sum_{j=1}^{n} W_{ij} \theta_{t,i,q_l}^{full}$
18:    $\theta_{t,i,q_{l+1}}^{full} \leftarrow \theta_{t,i,q_{l+\frac{1}{2}}}^{full} - \gamma\nabla\mathcal{L}_{cl}(\theta_{t,i,q_l}^{full}; \xi, \xi', \hat{\xi})$
19: **end for**
20:
21: **Function** UNLEARN:
22: **for** $q_f = 0$ to $Q_f - 1$ **do**
23:    **if** $q_f = 0$ **then**
24:       Initialize the unlearning module $\theta_i^{\text{ul}}$ from $\theta_i^{l}$
25:    **end if**
26:    Randomly sample $\hat{\xi}_f$ from $\mathcal{D}'_f$
27:    Compute the local stochastic gradient $\nabla\mathcal{L}_{ul}(\theta_i^e \diamond \theta_{i,q_f}^{\text{ul}}; \hat{\xi}_f)$            ▷ Eq. 11
28:    Update the unlearning module $\theta_{i,q_f+1}^{\text{ul}} \leftarrow \theta_{i,q_f}^{\text{ul}} - \gamma\nabla\mathcal{L}_{ul}(\theta_i^e \diamond \theta_{i,q_f}^{\text{ul}}; \hat{\xi}_f)$
29: **end for**
30: Update the classifier                                                       ▷ Eq. 12

---

- **Lander**: This represents a state-of-the-art approach in federated continual learning. Similar to Target, a central data generator is trained on the server using local models and can only generate unlabeled data. It should be noted that this method relies on the dataset-specific pre-trained label embedding to assist in training, which prevents it from being applicable to our first dataset *Grayscale*.

- **FedAF**: This method is inspired by the active forgetting mechanism in the human learning-memory system, and achieves unlearning by introducing noise to the target data. Specifically, the method adds noise to the labels of the target samples, while keeping the remaining data unchanged, and trains the model on the combined dataset to achieve forgetting. Moreover, this method does not require a recovery process; instead, it leverages the EWC method to maintain performance during unlearning. Therefore, we further integrate it with EWC to endow the model with continual learning capabilities. Due to the requirement of accessing the labeled data, it can only remove the latest learned knowledge acquired before the unlearning request.

- **EWC or LwF / AU**: FedAU involves training an auxiliary module on the target data distribution and achieves unlearning through a linear transformation. To enhance its ability to learn continually, we integrate this approach with EWC or LwF. Due to the requirement of accessing the data distribution, it can only remove the latest learned knowledge acquired before the unlearning request.

- **EWC or LwF / QDsga**: QuickDrop generates a distilled dataset using the cross-entropy loss during training, and achieves forgetting by applying Stochastic gradient ascent (SGA) on the target samples followed by a linear trans-

formation of the model weights. We combine it with EWC or LwF to achieve continual learning. Since its data synthesis process is designed only for a single learning task, we instead use stored real data to perform SGA. As a result, this method is capable of forgetting arbitrarily learned knowledge, but it does not comply with privacy-preserving requirements.

- **CLMUL**: This is the first framework that simultaneously considers continual learning and unlearning. However, it is implemented in a centralized setting and employs a buffer to store historical data for replay, which significantly differs from our DCU. We adapt it into a distributed formulation for comparison. This method is capable of forgetting arbitrarily learned knowledge. We set the buffer size of each client to 200.

- **Target or Lander / AU**: As previously discussed, AU enables unlearning with only access to the data distribution. This allows it to be integrated with Target or Lander, facilitating forgetting at arbitrary points in time.

## B.2. Topology Setting

Our experiments are conducted under dynamically changing network topologies. In each communication round, the existence of a connection between any pair of clients is randomly generated. Specifically, we construct an Erdős–Rényi random graph with an edge probability (density) of 0.6. Then, self-loops are explicitly enforced for all nodes to ensure that each client retains its own information during aggregation. Then, we use the Metropolis-Hastings method to generate the doubly stochastic mixing matrix as communication weights between clients

$$W_{ij} = \begin{cases} \frac{1}{\max(d_i, d_j)}, & \text{if } i \neq j \text{ and } (i,j) \in \mathcal{E} \\ 1 - \sum_{k \neq i} W_{ik}, & \text{if } i = j \\ 0, & \text{otherwise} \end{cases}$$

where $d_i$ denotes the degree of node $i$, and $\mathcal{E}$ is the set of edges in the communication graph.

## B.3. Client Setting

We set client number n = 10 for Grayscale, and 5 for RGB. The connections between clients are randomly generated in each round. For *Grayscale*, client employs a simple 2-layer CNN as the feature extractor for the classification model, and owns a pretrained grayscale diffusion model using con-minDiffusion [1]. For *RGB*, client owns a ResNet18 (He et al., 2016) and LDM [2] (Rombach et al., 2021). The hyperparameter $\beta_{cl}$ is set to 1 and 10 for the two datasets, respectively, while $\beta_{ul}$ is consistently set to 0.2 for both datasets.

## B.4. Configuration

*Grayscale*: For prototypes extraction, we configure 5 communication epoch and 50 local iteration steps with 128 batch size. Each client generates 200 synthetic samples as $\hat{\mathcal{D}}$, 100 samples as $\mathcal{D}'$ and 100 samples as $\mathcal{D}_f$ .For learning, we configure 10 global communication rounds per learning task and 1 local epoch. For unlearning, we configure 50 iteration steps with 64 batch size. The pre-trained diffusion model comes from the minDiffusion trained on Mnist, Emnist and Fashion-Mnist datasets after 100 epochs.
*RGB*: For prototypes extraction, we configure 10 communication epoch and 50 local iteration steps with 12 batch size. Each client generates 200 synthetic samples as $\hat{\mathcal{D}}$, 100 samples as $\mathcal{D}'$ and 100 samples as $\mathcal{D}_f$ .For learning, we configure 50 global communication rounds per learning task and 5 local epoch. For unlearning, we configure 50 iteration steps with 64 batch size.

We train the classifier with SGD and the learning rate is 0.01. We run all experiments on the GeForce RTX 3090 GPUs 24GB.

## B.5. Test setting

For the *Grayscale* setting, we utiliz the test sets of MNIST, EMNIST, and Fashion-MNIST as the evaluation datasets. For the *RGB* setting, we employ the corresponding CIFAR-100 test dataset.

---

[1]https://github.com/byrkbrk/conditional-ddpm
[2]https://github.com/CompVis/latent-diffusion

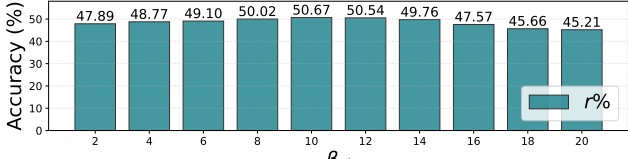 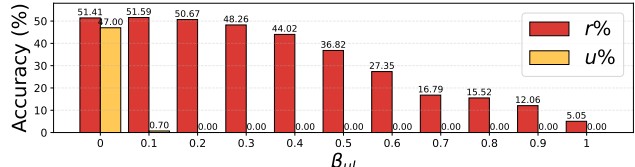

*Figure 4.* Hyperparameter analysis of $\beta_{cl}$ and $\beta_{ul}$.

*Table 5.* Speedup

| Methods | $r\uparrow/u\downarrow\%$ | Speedup |
|---------|---------------------------|---------|
| Retrain | 71.14(0.00) | $= 1$ |
| Ours | 50.67(0.00) | $\approx 0.006\times$ |

## C. Hyperparameter analysis

In DCU, $\beta_{cl}$ and $\beta_{ul}$ influence knowledge retention and knowledge forgetting, respectively. As shown in Figure 4, we conduct an analysis on the *RGB* dataset, measuring client average accuracy on both the retained and unlearned classes—denoted r% and u%, respectively.

For $\beta_{cl}$, results show that the default setting of $\beta_{cl} = 10$ achieves optimal performance with 50.67% accuracy on retained classes, while maintaining robust knowledge retention across the tested hyperparameter range of 2-20, despite slightly gradual performance degradation at both lower and higher values. For $\beta_{ul}$, the trend in the figure demonstrates that a critical threshold at 0.2, where 0% accuracy is achieved on unlearned classes while still maintaining 50.67% accuracy on retained knowledge. Below this value, unlearning remains incomplete, while higher values unnecessarily degrade model performance on remaining classes.

## D. Time Cost

This section compares the time cost of our method with naive retraining. We evaluate on *RGB*. Since retraining requires full data access, this method cannot support concurrent continual learning (CL) and unlearning (UL). Thus, we configure it under a standard UL setup: the retrain baseline essentially performs training on all data at once, with unlearning time equivalent to a single learning task on the remaining data (without class 0 and 20) with 50 global communication rounds and 5 local epochs per round. As shown in the Table 5, using the retrain time as the baseline, the average time for DCU to unlearn a single class is only 0.006 times that of retraining.

## E. Convergence analysis

**Assumption E.1.** (Lipschitzian gradient). *For every task $t \in [T]$, client $i \in [n]$ there exist a constant L, such that,*

$$\|\nabla f_{t,i}(\theta) - \nabla f_{t,i}(\theta')\| \leq L\|\theta - \theta'\|. \tag{13}$$

**Assumption E.2.** (Dynamic topology). *For every task, in each round $q_l$, the communication matrix $W_{q_l}$ is a real symmetric doubly stochastic matrix, satisfying $W_{ij} \in [0,1], \forall i, j, W_{ij} = W_{ji}, \forall i, j, \sum_j W_{ij} = 1, \forall i, \text{ and } W^q \mathbf{1}_n = \mathbf{1}_n$. We define*

$$\rho_q := (max\{|\lambda_2(W_q)|, |\lambda_n(W_q)|\})^2, \tag{14}$$

where $\lambda_l(\cdot)$ is the $l$-th largest eigenvalue of the matrix. And we write $\rho = \max_q \rho_q$.

**Assumption E.3.** (Bounded Local Gradient Difference). *For every task $t \in [T]$, client $i \in [n]$, and communication round $q_l \in [Q_l]$, there exists constants $\sigma_l$ such that the variance of stochastic gradients in client $i$ at task $t$ is bounded by*

$$\mathbb{E}\|\nabla F_{t,i}(\theta_{t,i,q_l}, \xi_{t,i,q_l}) - \nabla f_{t,i}(\theta_{t,i,q_l})\|^2 \leq \sigma_1^2. \tag{15}$$

$$\tag{16}$$

**Assumption E.4.** (Bounded Intra-task Gradient Difference). *For every task $t \in [T]$, client $i \in [n]$, and communication round $q_l \in [Q_l]$, there exists constants $\sigma_g$ such that the data heterogeneous in one task can be bounded by*

$$\|\nabla f_{t,i}(\theta_{t,i,q_l}) - \nabla f_t(\theta_{t,i,q_l})\|^2 \le \sigma_g^2. \tag{17}$$

**Assumption E.5.** (Bounded Inter-task Gradient Difference). *For every pair of tasks $t, t' \in [T]$, there exists constants $\sigma_t$ such that the difference of their global gradients can be bounded by:*

$$\|\nabla f_t(\theta) - \nabla f_{t'}(\theta)\|^2 \le \sigma_t^2. \tag{18}$$

To view the algorithm from a global view, we define $\Theta_{t,q_l} := [\theta_{t,1,q_l} \cdots \theta_{t,n,q_l}]$ as the concatenation of all trained parameters, $\partial f(\Theta_{t,q_l}) := [\nabla f_1(\theta_{t,1,q_l}) \cdots \nabla f_n(_{t,n,q_l})]$ as the concatenation of all gradient of hypernetwork. Then, we can gain the following convergence result. And the whole optimization problem for distributed continual leraning is:

$$\min f_{1:T}(\theta_T) = \frac{1}{n} \sum_{t=1}^{T} \sum_{i=1}^{n} f_{t,i}(\theta_T) \tag{19}$$

**Theorem E.6.** *Under Assumptions 1-5, let $\Phi_1 = 1 - \frac{18\gamma^2 nL^2}{(1-\sqrt{\rho})^2}$, $\Delta_f = \mathbb{E}f_{1:T}\left(\frac{\Theta_{T,0}\mathbf{1}_n}{n}\right) - \mathbb{E}f_{1:T}\left(\frac{\Theta_{T,Q_l}\mathbf{1}_n}{n}\right)$, we have the following convergence rate for our proposed algorithm:*

$$\frac{1}{Q_l} \sum_{q_l=0}^{Q_l-1} \mathbb{E}\|\nabla f_{1:T}\left(\frac{\Theta_{T,q_l}\mathbf{1}_n}{n}\right)\|^2$$

$$\le \frac{2\Delta_f}{\gamma Q_l} + \mathbb{E}\|\nabla f_{1:T-1}\left(\frac{\Theta_{T,0}\mathbf{1}_n}{n}\right)\|^2$$

$$+ \left((T-1) + \frac{18\gamma^2(T+1)L^2 n}{(1-\sqrt{\rho})^2\Phi_1}\right)\frac{1}{Q_l} \sum_{q_l=0}^{Q_l-1} \mathbb{E}\|\nabla f_T\left(\frac{\Theta_{T,q_l}\mathbf{1}_n}{n}\right)\|^2$$

$$+ \gamma L \frac{1}{Q_l} \sum_{q_l=0}^{Q_l-1} \mathbb{E}\|\frac{\partial f_T(\Theta_{T,q_l})\mathbf{1}_n}{n}\|^2 + \frac{\gamma L}{n}(1 + \gamma L(T-1)^2)\sigma_l^2$$

$$+ T^2\sigma_t^2 + \frac{2\gamma^2(T+1)L^2 n}{(1-\rho)\Phi_1}\sigma_l^2 + \frac{18\gamma^2(T+1)L^2 n}{(1-\sqrt{\rho})^2\Phi_1}\sigma_g^2$$

**Corollary E.7.** *If we choose the learning rate $\gamma = \sqrt{\frac{n}{Q_l}}$, let the total number of iterate $Q_l \ge \frac{36L^2 n^2}{(1-\sqrt{\rho})^2}$ is sufficiently large. Denote $\mathbb{E}\|\nabla f_{1:T-1}\left(\frac{\Theta_{T,0}\mathbf{1}_n}{n}\right)\|^2 \le C$, where $C$ is a constant, $T^2\sigma_t^2$ is also a constant corresponding to the task. We can get the following convergence rate:*

$$\frac{1}{Q_l} \sum_{q_l=0}^{Q_l-1} \mathbb{E}\|\nabla f_{1:T}\left(\frac{\Theta_{T,q_l}\mathbf{1}_n}{n}\right)\|^2 \le \mathcal{O}(\frac{1}{\sqrt{nQ_l}} + \frac{1}{Q_l})$$

**Lemma E.8.** *Under Assumption E.2, for all $i \in [n]$*

$$\|\frac{\mathbf{1}_n}{n} - W_q W_{q-1} W_{q-2} \cdots W_l e_i\|^2 \le \rho^{q-l-1}. \tag{20}$$

*Proof to Lemma 1.*

$$\|\frac{\mathbf{1}_n}{n} - W_q W_{q-1} \cdots W_l e_i\|^2$$

$$\le \rho_q \|\frac{\mathbf{1}_n}{n} - W_{q-1} \cdots W_l e_i\|^2$$

$$\le \rho_q \rho_{q-1} \|\frac{\mathbf{1}_n}{n} - W_{q-2} \cdots W_l e_i\|^2$$

$$\vdots$$

$$\leq \rho_q \rho_{q-1} \cdots \rho_{l+1} \left\| \frac{\mathbf{1}_n}{n} - W_l e_i \right\|^2$$
$$\leq \rho_q \rho_{q-1} \cdots \rho_{l+1} \rho_l$$
$$\leq \rho^{q-l+1}$$

$$\mathbb{E} f_{1:T} \left( \frac{\Theta_{T,q_l+1} \mathbf{1}_n}{n} \right)$$
$$= \mathbb{E} f_{1:T} \left( \frac{\Theta_{T,q_l} \mathbf{1}_n}{n} - \gamma \frac{\partial f_T(\Theta_{T,q_l}, \xi_{T,q_l}) \mathbf{1}_n}{n} \right)$$
$$\leq \mathbb{E} f_{1:T} \left( \frac{\Theta_{T,q_l} \mathbf{1}_n}{n} \right) + \mathbb{E} \left\langle \nabla f_{1:T} \left( \frac{\Theta_{T,q_l} \mathbf{1}_n}{n} \right), \frac{-\gamma \partial f_T(\Theta_{T,q_l}) \mathbf{1}_n}{n} \right\rangle$$
$$+ \frac{\gamma^2 L}{2} \mathbb{E} \left\| \frac{\sum_{i=1}^n \nabla F_{T,i}(\theta_{T,i,q_l}, \xi_{T,i,q_l})}{n} \right\|^2$$
$$= \mathbb{E} f_{1:T} \left( \frac{\Theta_{T,q_l} \mathbf{1}_n}{n} \right) + \mathbb{E} \left\langle \nabla f_{1:T} \left( \frac{\Theta_{T,q_l} \mathbf{1}_n}{n} \right), \frac{-\gamma \partial f_T(\Theta_{T,q_l}) \mathbf{1}_n}{n} \right\rangle$$
$$+ \frac{\gamma^2 L}{2} \mathbb{E} \left\| \frac{\sum_{i=1}^n \nabla F_{T,i}(\theta_{T,i,q_l}, \xi_{T,i,q_l})}{n} - \frac{\sum_{i=1}^n \nabla f_{T,i}(\theta_{T,i,q_l})}{n} \right\|^2$$
$$+ \frac{\gamma^2 L}{2} \mathbb{E} \left\| \frac{\sum_{i=1}^n \nabla f_{T,i}(\theta_{T,i,q_l})}{n} \right\|^2$$
$$\leq \mathbb{E} f_{1:T} \left( \frac{\Theta_{T,q_l} \mathbf{1}_n}{n} \right) + \underbrace{\mathbb{E} \left\langle \nabla f_{1:T} \left( \frac{\Theta_{T,q_l} \mathbf{1}_n}{n} \right), \frac{-\gamma \partial f_T(\Theta_{T,q_l}) \mathbf{1}_n}{n} \right\rangle}_{T_1}$$
$$+ \frac{\gamma^2 L}{2n} \sigma_l^2 + \frac{\gamma^2 L}{2} \mathbb{E} \left\| \frac{\partial f_T(\Theta_{T,q_l}) \mathbf{1}_n}{n} \right\|^2$$

To bound $T_1$:

$$T_1 = \mathbb{E} \left\langle \nabla f_{1:T} \left( \frac{\Theta_{T,q_l} \mathbf{1}_n}{n} \right), \frac{-\gamma \partial f_T(\Theta_{T,q_l}) \mathbf{1}_n}{n} \right\rangle$$
$$= \mathbb{E} \left\langle \nabla f_{1:T} \left( \frac{\Theta_{T,q_l} \mathbf{1}_n}{n} \right), -\frac{\gamma \partial f_T(\Theta_{T,q_l}) \mathbf{1}_n}{n} \right.$$
$$\left. -\gamma \nabla f_T \left( \frac{\Theta_{T,q_l} \mathbf{1}_n}{n} \right) + \gamma \nabla f_T \left( \frac{\Theta_{T,q_l} \mathbf{1}_n}{n} \right) \right\rangle$$
$$= \mathbb{E} \left\langle \nabla f_{1:T} \left( \frac{\Theta_{T,q_l} \mathbf{1}_n}{n} \right), -\frac{\gamma \partial f_T(\Theta_{T,q_l}) \mathbf{1}_n}{n} + \gamma \nabla f_T \left( \frac{\Theta_{T,q_l} \mathbf{1}_n}{n} \right) \right\rangle$$
$$+ \mathbb{E} \left\langle \nabla f_{1:T} \left( \frac{\Theta_{T,q_l} \mathbf{1}_n}{n} \right), -\gamma \nabla f_T \left( \frac{\Theta_{T,q_l} \mathbf{1}_n}{n} \right) \right\rangle$$
$$= \underbrace{\mathbb{E} \left\langle \nabla f_{1:T} \left( \frac{\Theta_{T,q_l} \mathbf{1}_n}{n} \right), -\frac{\gamma \partial f_T(\Theta_{T,q_l}) \mathbf{1}_n}{n} + \gamma \nabla f_T \left( \frac{\Theta_{T,q_l} \mathbf{1}_n}{n} \right) \right\rangle}_{T_2}$$
$$+ \underbrace{\mathbb{E} \left\langle \nabla f_{1:T-1} \left( \frac{\Theta_{T,q_l} \mathbf{1}_n}{n} \right), -\gamma \nabla f_T \left( \frac{\Theta_{T,q_l} \mathbf{1}_n}{n} \right) \right\rangle}_{T_3}$$
$$- \gamma \mathbb{E} \left\| \nabla f_T \left( \frac{\Theta_{T,q_l} \mathbf{1}_n}{n} \right) \right\|^2$$

To bound $T_2$, we substitute $\nabla f_{1:T} = \sum_{t=1}^T \nabla f_t = T \nabla f_T + \sum_{t=1}^T (\nabla f_t - \nabla f_T)$.

$$T_2 = \gamma \mathbb{E} \left\langle \nabla f_{1:T} \left( \frac{\Theta_{T,q_l} \mathbf{1}_n}{n} \right), \right.$$

$$\nabla f_T\left(\frac{\Theta_{T,q_l}\mathbf{1}_n}{n}\right) - \frac{\partial f_T(\Theta_{T,q_l})\mathbf{1}_n}{n}\Big\rangle$$

$$= \gamma\mathbb{E}\left\langle T\nabla f_T\left(\frac{\Theta_{T,q_l}\mathbf{1}_n}{n}\right)\right.$$

$$+ \sum_{t=1}^{T}\left(\nabla f_t\left(\frac{\Theta_{T,q_l}\mathbf{1}_n}{n}\right) - \nabla f_T\left(\frac{\Theta_{T,q_l}\mathbf{1}_n}{n}\right)\right),$$

$$\nabla f_T\left(\frac{\Theta_{T,q_l}\mathbf{1}_n}{n}\right) - \frac{\partial f_T(\Theta_{T,q_l})\mathbf{1}_n}{n}\Big\rangle$$

$$= \gamma T\mathbb{E}\langle\nabla f_T, \nabla f_T\left(\frac{\Theta_{T,q_l}\mathbf{1}_n}{n}\right) - \frac{\partial f_T(\Theta_{T,q_l})\mathbf{1}_n}{n}\rangle$$

$$+ \gamma\mathbb{E}\left\langle\sum_{t=1}^{T}(\nabla f_t - \nabla f_T), \nabla f_T\left(\frac{\Theta_{T,q_l}\mathbf{1}_n}{n}\right) - \frac{\partial f_T(\Theta_{T,q_l})\mathbf{1}_n}{n}\right\rangle$$

$$\leq \frac{\gamma T}{2}\mathbb{E}\|\nabla f_T\|^2$$

$$+ \frac{\gamma T}{2}\mathbb{E}\|\nabla f_T\left(\frac{\Theta_{T,q_l}\mathbf{1}_n}{n}\right) - \frac{\partial f_T(\Theta_{T,q_l})\mathbf{1}_n}{n}\|^2$$

$$+ \frac{\gamma}{2}\mathbb{E}\|\sum_{t=1}^{T}(\nabla f_t - \nabla f_T)\|^2$$

$$+ \frac{\gamma}{2}\mathbb{E}\|\nabla f_T\left(\frac{\Theta_{T,q_l}\mathbf{1}_n}{n}\right) - \frac{\partial f_T(\Theta_{T,q_l})\mathbf{1}_n}{n}\|^2$$

$$\leq \frac{\gamma T}{2}\mathbb{E}\|\nabla f_T\|^2$$

$$+ \frac{\gamma(T+1)}{2}\underbrace{\mathbb{E}\|\nabla f_T\left(\frac{\Theta_{T,q_l}\mathbf{1}_n}{n}\right) - \frac{\partial f_T(\Theta_{T,q_l})\mathbf{1}_n}{n}\|^2}_{T_{21}}$$

$$+ \frac{\gamma T}{2}\sum_{t=1}^{T}\mathbb{E}\|\nabla f_t - \nabla f_T\|^2$$

$$\leq \frac{\gamma T}{2}\mathbb{E}\|\nabla f_T\left(\frac{\Theta_{T,q_l}\mathbf{1}_n}{n}\right)\|^2 + \frac{\gamma(T+1)}{2}T_{21} + \frac{\gamma T^2}{2}\sigma_t^2$$

To bound $T_{21}$:

$$T_{21} = \mathbb{E}\|\nabla f_T\left(\frac{\Theta_{T,q_l}\mathbf{1}_n}{n}\right) - \frac{\partial f_T(\Theta_{T,q_l})\mathbf{1}_n}{n}\|^2$$

$$\leq \frac{1}{n}\sum_{i=1}^{n}\mathbb{E}\|\nabla f_{T,i}\left(\sum_{i'=1}^{n}\frac{\theta_{T,i',q_l}}{n}\right) - \nabla f_{T,i}(\theta_{T,i,q_l})\|^2$$

$$\leq \frac{L^2}{n}\sum_{i=1}^{n}\underbrace{\mathbb{E}\|\frac{\sum_{i'=1}^{n}\theta_{T,i',q_l}}{n} - \theta_{T,i,q_l}\|^2}_{:=Q_{q_l,i}}$$

**Analysis of $Q_{q_l,i}$:**

$$Q_{q_l,i} = \mathbb{E}\|\frac{\sum_{i'=1}^{n}\theta_{T,i',q_l}}{n} - \theta_{T,i,q_l}\|^2$$

$$= \mathbb{E}\|\frac{\Theta_{T,q_l}\mathbf{1}_n}{n} - \Theta_{T,q_l}e_i\|^2$$

$$= \mathbb{E}\|\frac{\Theta_{T,q_l-1}W_{q_l-1}\mathbf{1}_n - \gamma\partial f_T(\Theta_{T,q_l-1},\xi_{T,q_l-1})\mathbf{1}_n}{n}$$

$$- (\Theta_{T,q_l-1}W_{q_l-1}e_i - \gamma\partial f_T(\Theta_{T,q_l-1},\xi_{T,q_l-1})e_i)\|^2$$

$$= \mathbb{E}\|\frac{\Theta_{T,q_l-1}\mathbf{1}_n - \gamma\partial f_T(\Theta_{T,q_l-1}, \xi_{T,q_l-1})\mathbf{1}_n}{n}$$

$$- (\Theta_{T,q_l-1}W_{q_l-1}e_i - \gamma\partial f_T(\Theta_{T,q_l-1}, \xi_{T,q_l-1})e_i)\|^2$$

$$= \mathbb{E}\|\frac{\Theta_{T,0}\mathbf{1}_n - \gamma\sum_{q=0}^{q_l-1}\partial f_T(\Theta_{T,q}, \xi_{T,q})\mathbf{1}_n}{n}$$

$$- (\Theta_{T,0}W_{q_l-1}\cdots W_0 e_i$$

$$- \sum_{q'=0}^{q_l-1}\gamma\partial f_T(\Theta_{T,q'}, \xi_{T,q'})W_{q_l-2}\cdots W_{q'}e_i)\|^2$$

$$= \mathbb{E}\|\Theta_{T,0}\left(\frac{\mathbf{1}_n}{n} - W_{q_l-1}\cdots W_0 e_i\right)$$

$$- \sum_{q'=0}^{q_l-1}\gamma\cdot\partial F_T(\Theta_{T,q'}, \xi_{T,q'})\cdot\left(\frac{\mathbf{1}_n}{n} - W_{q_l-2}\cdots W_{q'}e_i\right)\|^2$$

$$= \mathbb{E}\|\sum_{q'=0}^{q_l-1}\gamma\cdot\partial F_T(\Theta_{T,q'}, \xi_{T,q'})\left(\frac{\mathbf{1}_n}{n} - W_{q_l-2}\cdots W_{q'}e_i\right)\|^2$$

$$= \gamma^2\mathbb{E}\|\sum_{q'=0}^{q_l-1}\partial F_T(\Theta_{T,q'}, \xi_{T,q'})\left(\frac{\mathbf{1}_n}{n} - W_{q_l-2}\cdots W_{q'}e_i\right)\|^2$$

$$= \gamma^2\mathbb{E}\|\sum_{q'=0}^{q_l-1}(\partial F_T(\Theta_{T,q'}, \xi_{T,q'}) - \partial f_T(\Theta_{T,q'})$$

$$+ \partial f_T(\Theta_{T,q'}))\left(\frac{\mathbf{1}_n}{n} - W_{q_l-2}\cdots W_{q'}e_i\right)\|^2$$

$$\leq 2\gamma^2\mathbb{E}\|\sum_{q'=0}^{q_l-1}(\partial F_T(\Theta_{T,q'}, \xi_{T,q'})$$

$$- \partial f_T(\Theta_{T,q'})\left(\frac{\mathbf{1}_n}{n} - W_{q_l-2}\cdots W_{q'}e_i\right)\|^2$$

$$+ 2\gamma^2\mathbb{E}\|\sum_{q'=0}^{q_l-1}\partial f_T(\Theta_{T,q'})\left(\frac{\mathbf{1}_n}{n} - W_{q_l-2}\cdots W_{q'}e_i\right)\|^2$$

$$\leq 2n\gamma^2\sigma_l^2\mathbb{E}\|\frac{\mathbf{1}_n}{n} - W_{q_l-2}\cdots W_{q'}e_i\|^2$$

$$+ 2\gamma^2\underbrace{\mathbb{E}\|\sum_{q'=0}^{q_l-1}\partial f_T(\Theta_{T,q'})\left(\frac{\mathbf{1}_n}{n} - W_{q_l-2}\cdots W_{q'}e_i\right)\|^2}_{T_{q1}}$$

$$\leq \frac{2n\gamma^2\sigma_l^2}{1-\rho} + 2\gamma^2 T_{q1}$$

To bound $T_{q1}$

$$\mathbb{E}\|\sum_{q'=0}^{q_l-1}\partial f_T(\Theta_{T,q'})\left(\frac{\mathbf{1}_n}{n} - W_{q_l-2}\cdots W_{q'}e_i\right)\|^2$$

$$= \underbrace{\sum_{q'=0}^{q_l-1}\mathbb{E}\|\partial f_T(\Theta_{T,q'})\left(\frac{\mathbf{1}_n}{n} - W_{q_l-2}\cdots W_{q'}e_i\right)\|^2}_{T_{q2}}$$

$$+ \underbrace{\sum_{q' \neq \tilde{q}}^{q_l - 1} \mathbb{E} \langle \partial f_T(\Theta_{T,q'}) \left( \frac{\mathbf{1}_n}{n} - W_{q_l-2} \cdots W_{q'} e_i \right),}_{T_{q3}}$$

$$\underbrace{\partial f_T(\Theta_{T,\tilde{q}}) \left( \frac{\mathbf{1}_n}{n} - W_{q_l-2} \cdots W_{\tilde{q}} e_i \right) \rangle}_{T_{q3}}$$

To bound $T_{q2}$:

$$T_{q2} \leq \sum_{q'=0}^{q_l-1} \mathbb{E} \| \partial f_T(\Theta_{T,q'}) \| \| \frac{\mathbf{1}_n}{n} - W_{q_l-2} \cdots W_{q'} e_i \|^2$$

$$\leq \sum_{q'=0}^{q_l-1} 3\mathbb{E} \| \partial f_T(\Theta_{T,q'})$$

$$- \partial f_T \left( \frac{\Theta_{T,q'} \mathbf{1}_n \mathbf{1}_n^T}{n} \right) \|^2 \| \frac{\mathbf{1}_n}{n} - W_{q_l-2} \cdots W_{q'} e_i \|^2$$

$$+ \sum_{q'=0}^{q_l-1} 3\mathbb{E} \| \partial f_T \left( \frac{\Theta_{T,q'} \mathbf{1}_n \mathbf{1}_n^T}{n} \right)$$

$$- \nabla f_T \left( \frac{\Theta_{T,q'} \mathbf{1}_n}{n} \right) \mathbf{1}_n^T \|^2 \| \frac{\mathbf{1}_n}{n} - W_{q_l-2} \cdots W_{q'} e_i \|^2$$

$$+ \sum_{q'=0}^{q_l-1} 3\mathbb{E} \| \nabla f_T \left( \frac{\Theta_{T,q'} \mathbf{1}_n}{n} \right) \mathbf{1}_n^T \|^2 \| \frac{\mathbf{1}_n}{n} - W_{q_l-2} \cdots W_{q'} e_i \|^2$$

$$\leq 3L^2 \sum_{q'=0}^{q_l-1} \sum_{h=1}^{n} \mathbb{E} Q_{q',h} \| \frac{\mathbf{1}_n}{n} - W_{q_l-2} \cdots W_{q'} e_i \|^2$$

$$+ 3n\sigma_g^2 \cdot \frac{1}{1-\rho}$$

$$+ 3 \sum_{q'=0}^{q_l-1} \mathbb{E} \| \nabla F \left( \frac{\Theta_{T,q'} \mathbf{1}_n}{n} \right) \mathbf{1}_n^T \|^2 \| \frac{\mathbf{1}_n}{n} - W_{q_l-2} \cdots W_{q'} e_i \|^2$$

To bound $T_{q3}$:

$$T_{q3} = \sum_{q' \neq \tilde{q}}^{q_l-1} \mathbb{E} \langle \partial f_T(\Theta_{T,q'}) \left( \frac{\mathbf{1}_n}{n} - W_{q_l-2} \cdots W_{\tilde{q}} e_i \right)$$

$$, \partial f_T(\Theta_{T,\tilde{q}}) \left( \frac{\mathbf{1}_n}{n} - W_{q_l-2} \cdots W_{\tilde{q}} e_i \right) \rangle$$

$$\leq \sum_{q' \neq \tilde{q}}^{q_l-1} \mathbb{E} \| \partial f_T(\Theta_{T,q'}) \left( \frac{\mathbf{1}_n}{n} - W_{q_l-2} \cdots W_{q'} e_i \right) \|$$

$$\cdot \| \partial f_T(\Theta_{T,\tilde{q}}) \left( \frac{\mathbf{1}_n}{n} - W_{q_l-2} \cdots W_{\tilde{q}} e_i \right) \|$$

$$\leq \sum_{q' \neq \tilde{q}}^{q_l-1} \mathbb{E} \| \partial f_T(\Theta_{T,q'}) \| \| \frac{\mathbf{1}_n}{n} - W_{q_l-2} \cdots W_{q'} e_i \|$$

$$\cdot \| \partial f_T(\Theta_{T,\tilde{q}}) \| \| \frac{\mathbf{1}_n}{n} - W_{q_l-2} \cdots W_{\tilde{q}} e_i \|$$

$$\leq \sum_{q' \neq \tilde{q}}^{q_l - 1} \mathbb{E} \frac{\|\partial f_T(\Theta_{T,q'})\|^2}{2} \|\frac{\mathbf{1}_n}{n} - W_{q_l - 2} \cdots W_{q'} e_i\| \|\frac{\mathbf{1}_n}{n} - W_{q_l - 2} \cdots W_{\tilde{q}} e_i\|$$

$$+ \sum_{q' \neq \tilde{q}}^{q_l - 1} \mathbb{E} \frac{\|\partial f_T(\Theta_{T,\tilde{q}})\|^2}{2} \|\frac{\mathbf{1}_n}{n} - W_{q_l - 2} \cdots W_{q'} e_i\| \|\frac{\mathbf{1}_n}{n} - W_{q_l - 2} \cdots W_{\tilde{q}} e_i\|$$

$$\leq \sum_{q' \neq \tilde{q}}^{q_l - 1} \mathbb{E} \left( \frac{\|\partial f_T(\Theta_{T,q'})\|}{2} + \frac{\|\partial f_T(\Theta_{T,\tilde{q}})\|^2}{2} \right) \cdot \rho^{q_l - \frac{q' + \tilde{q}}{2} - 1}$$

$$\leq \sum_{q' \neq \tilde{q}}^{q_l - 1} \mathbb{E} \left( \|\partial f_T(\Theta_{T,q'})\|^2 \right) \cdot \rho^{q_l - \frac{q' + \tilde{q}}{2} - 1}$$

$$\leq 3 \underbrace{\sum_{q' \neq \tilde{q}}^{q_l - 1} \left( \sum_{h=1}^{n} \mathbb{E} L^2 \mathbb{E} Q_{q',h} + \mathbb{E} \|\nabla f_T(\frac{\Theta_{T,q'} \mathbf{1}_n}{n}) \mathbf{1}_n^T\| \right) \cdot \rho^{q_l - \frac{q' + \tilde{q}}{2} - 1}}_{T_{q4}}$$

$$+ \underbrace{\sum_{q' \neq \tilde{q}}^{q_l - 1} 3 n \sigma_g^2 \rho^{q_l - \frac{q' + \tilde{q}}{2} - 1}}_{T_{q5}}$$

To bound $T_{q4}$:

$$T_{q4} = 6 \sum_{q'=0}^{q_l - 1} \left( \sum_{h=1}^{n} \mathbb{E} L^2 \mathbb{E} Q_{q',h} + \mathbb{E} \|\nabla f_T(\frac{\Theta_{T,q'} \mathbf{1}_n}{n}) \cdot \mathbf{1}_n^T\| \right)$$

$$\cdot \sum_{p=q'+1}^{q_l - 1} \sqrt{\rho}^{2q_l - q' - p - 2}$$

$$\leq 6 \sum_{q'=0}^{q_l - 1} \left( \sum_{h=1}^{n} \mathbb{E} L^2 \mathbb{E} Q_{q',h} + \mathbb{E} \|\nabla f_T(\frac{\Theta_{T,q'} \mathbf{1}_n}{n}) \cdot \mathbf{1}_n^T\| \right) \cdot \frac{\sqrt{\rho}^{q_l - q' - 1}}{1 - \sqrt{\rho}}$$

To bound $T_{q5}$:

$$T_{q5} \leq 6 n \sigma_g^2 \sum_{p > \tilde{q}}^{q_l - 1} \rho^{q_l - \frac{q' + \tilde{q}}{2} - 1}$$

$$= 6 n \sigma_g^2 \frac{(\rho^{q_l/2} - 1)(\rho^{q_l/2} - \sqrt{\rho})}{(\sqrt{\rho} - 1)^2 (\sqrt{\rho} + 1)}$$

$$\leq 6 n \sigma_g^2 \frac{1}{(1 - \sqrt{\rho})^2}$$

Plugging $T_{q4}$, $T_{q5}$ into $T_{q3}$:

$$T_{q3} \leq 6 \sum_{q'=0}^{q_l - 1} \left( \sum_{h=1}^{n} \mathbb{E} L^2 \mathbb{E} Q_{q',h} + \mathbb{E} \|\nabla f_T(\frac{\Theta_{T,q'} \mathbf{1}_n}{n}) \right.$$

$$\left. \cdot \mathbf{1}_n^T\| \right) \cdot \frac{\sqrt{\rho}^{q_l - q' - 1}}{1 - \sqrt{\rho}} + 6 n \sigma_g^2 \frac{1}{(1 - \sqrt{\rho})^2}$$

Plugging $T_{q3}, T_{q2}$ into $T_{q1}$:

$$
\begin{aligned}
T_{q1} &\leq 3L^2 \sum_{q'=0}^{q_l-1} \sum_{h=1}^{n} \mathbb{E}Q_{q',h} \|\frac{\mathbf{1}_n}{n} - W_{q_l-2} \cdots W_{q'} e_i\|^2 \\
&+ 3n\sigma_g^2 \cdot \frac{1}{1-\rho} \\
&+ 3 \sum_{q'=0}^{q_l-1} \mathbb{E}\|\nabla f_T \left(\frac{\Theta_{T,q'}\mathbf{1}_n}{n}\right) \mathbf{1}_n^T\|^2 \|\frac{\mathbf{1}_n}{n} - W_{q_l-2} \cdots W_{q'} e_i\|^2 \\
&+ 6 \sum_{q'=0}^{q_l-1} \left( \sum_{h=1}^{n} \mathbb{E}L^2 \mathbb{E}Q_{q',h} + \mathbb{E}\|\nabla f_T(\frac{\Theta_{T,q'}\mathbf{1}_n}{n}) \cdot \mathbf{1}_n^T\| \right) \cdot \frac{\sqrt{\rho}^{q_l-q'-1}}{1-\sqrt{\rho}} \\
&+ 6n\sigma_g^2 \frac{1}{(1-\sqrt{\rho})^2} \\
&\leq 3L^2 \sum_{q'=0}^{q_l-1} \sum_{h=1}^{n} \mathbb{E}Q_{q',h} \|\frac{\mathbf{1}_n}{n} - W_{q_l-2} \cdots W_{q'} e_i\|^2 \\
&+ 3 \sum_{q'=0}^{q_l-1} \mathbb{E}\|\nabla f_T \left(\frac{\Theta_{T,q'}\mathbf{1}_n}{n}\right) \mathbf{1}_n^T\|^2 \|\frac{\mathbf{1}_n}{n} - W_{q_l-2} \cdots W_{q'} e_i\|^2 \\
&+ 6 \sum_{q'=0}^{q_l-1} \left( \sum_{h=1}^{n} \mathbb{E}L^2 \mathbb{E}Q_{q',h} + \mathbb{E}\|\nabla f_T(\frac{\Theta_{T,q'}\mathbf{1}_n}{n}) \cdot \mathbf{1}_n^T\| \right) \\
&\cdot \frac{\sqrt{\rho}^{q_l-q'-1}}{1-\sqrt{\rho}} + \frac{9n\sigma_g^2}{(1-\sqrt{\rho})^2}
\end{aligned}
$$

Plugging $T_{q1}$ to $Q_{q_l,i}$:

$$
\begin{aligned}
Q_{q_l,i} &\leq \frac{2n\gamma^2\sigma_l^2}{1-\rho} + 6\gamma^2 L^2 \sum_{q'=0}^{q_l-1} \sum_{h=1}^{n} \mathbb{E}Q_{q',h} \|\frac{\mathbf{1}_n}{n} - W_{q_l-2} \cdots W_{q'} e_i\|^2 \\
&+ 6\gamma^2 \sum_{q'=0}^{q_l-1} \mathbb{E}\|\nabla f_T \left(\frac{\Theta_{T,q'}\mathbf{1}_n}{n}\right) \mathbf{1}_n^T\|^2 \|\frac{\mathbf{1}_n}{n} - W_{q_l-2} \cdots W_{q'} e_i\|^2 \\
&+ 12\gamma^2 \sum_{q'=0}^{q_l-1} \left( \sum_{h=1}^{n} \mathbb{E}L^2 \mathbb{E}Q_{q',h} + \mathbb{E}\|\nabla f_T(\frac{\Theta_{T,q'}\mathbf{1}_n}{n}) \cdot \mathbf{1}_n^T\| \right) \\
&\cdot \frac{\sqrt{\rho}^{q_l-q'-1}}{1-\sqrt{\rho}} + \frac{18\gamma^2 n\sigma_g^2}{(1-\sqrt{\rho})^2} \\
&\leq \frac{2n\gamma^2\sigma_l^2}{1-\rho} + 6\gamma^2 L^2 \sum_{q'=0}^{q_l-1} \sum_{h=1}^{n} \mathbb{E}Q_{q',h} \rho^{q_l-q'-1} \\
&+ 6\gamma^2 \sum_{q'=0}^{q_l-1} \mathbb{E}\|\nabla f_T(\frac{\Theta_{T,q'}\mathbf{1}_n}{n}) \cdot \mathbf{1}_n^T\|^2 \cdot \rho^{q_l-q'-1} \\
&+ 12\gamma^2 \sum_{q'=0}^{q_l-1} \left( \sum_{h=1}^{n} \mathbb{E}L^2 Q_{q',h} + \mathbb{E}\|\nabla f_T(\frac{\Theta_{T,q'}\mathbf{1}_n}{n}) \cdot \mathbf{1}_n^T\| \right) \\
&\cdot \frac{\sqrt{\rho}^{q_l-q'-1}}{1-\sqrt{\rho}} + \frac{18\gamma^2 n\sigma_g^2}{(1-\sqrt{\rho})^2} \\
&= \frac{2n\gamma^2\sigma_l^2}{1-\rho} + \frac{18\gamma^2 n\sigma_g^2}{(1-\sqrt{\rho})^2}
\end{aligned}
$$

$$+ 6\gamma^2 \sum_{q'=0}^{q_l-1} \mathbb{E}\|\nabla f_T(\frac{\Theta_{T,q'}\mathbf{1}_n}{n}) \cdot \mathbf{1}_n^T\|^2 \left(\rho^{q_l-q'-1} + \frac{2\sqrt{\rho}^{q_l-q'-1}}{1-\sqrt{\rho}}\right)$$

$$+ 6\gamma^2 L_\theta^2 \sum_{q'=0}^{q_l-1} \sum_{h=1}^{n} \mathbb{E}Q_{q',h} \left(\rho^{q_l-q'-1} + \frac{2\sqrt{\rho}^{q_l-q'-1}}{1-\sqrt{\rho}}\right)$$

Average $Q_{q_l}$ on all clients, we can get,

$$\mathbb{E}M_{q_l} = \frac{\mathbb{E}\sum_{i=1}^{n} Q_{q_l,i}}{n}$$

$$\leq \frac{2n\gamma^2\sigma_l^2}{1-\rho} + \frac{18\gamma^2 n\sigma_g^2}{(1-\sqrt{\rho})^2}$$

$$+ 6\gamma^2 \sum_{q'=0}^{q_l-1} \mathbb{E}\|\nabla f_T(\frac{\Theta_{T,q'}\mathbf{1}_n}{n}) \cdot \mathbf{1}_n^T\|^2 \left(\rho^{q_l-q'-1} + \frac{2\sqrt{\rho}^{q_l-q'-1}}{1-\sqrt{\rho}}\right)$$

$$+ 6\gamma^2 n L^2 \sum_{q'=0}^{q_l-1} \mathbb{E}M_{q'} \left(\rho^{q_l-q'-1} + \frac{2\sqrt{\rho}^{q_l-q'-1}}{1-\sqrt{\rho}}\right)$$

Summing from $q_l = 0$ to $Q_l$ we get:

$$\sum_{q_l=0}^{Q_l-1} \mathbb{E}M_{q_l} \leq \frac{2nQ_l\gamma^2\sigma_l^2}{1-\rho}$$

$$+ \frac{18\gamma^2 n\sigma_g^2 Q_l}{(1-\sqrt{\rho})^2} + 6\gamma^2 \sum_{q_l=0}^{Q_l-1}\sum_{q'=0}^{q_l-1} \mathbb{E}\|\nabla f_T(\frac{\Theta_{T,q'}\mathbf{1}_n}{n})$$

$$\cdot \mathbf{1}_n^T\|^2 \left(\rho^{q_l-q'-1} + \frac{2\sqrt{\rho}^{q_l-q'-1}}{1-\sqrt{\rho}}\right)$$

$$+ 6\gamma^2 n L^2 \sum_{q_l=0}^{Q_l-1}\sum_{q'=0}^{q_l-1} \mathbb{E}M_{q'} \left(\rho^{q_l-q'-1} + \frac{2\sqrt{\rho}^{q_l-q'-1}}{1-\sqrt{\rho}}\right)$$

$$\leq \frac{2nQ_l\gamma^2\sigma_l^2}{1-\rho} + \frac{18\gamma^2 n\sigma_g^2 Q_l}{(1-\sqrt{\rho})^2}$$

$$+ 6\gamma^2 \sum_{q_l=0}^{Q_l-1} \mathbb{E}\|\nabla f_T(\frac{\Theta_{T,q_l}\mathbf{1}_n}{n})$$

$$\cdot \mathbf{1}_n^T\|^2 \left(\sum_{i=0}^{\infty}\rho^i + \frac{2\sum_{i=0}^{\infty}\sqrt{\rho}^i}{1-\sqrt{\rho}}\right)$$

$$+ 6\gamma^2 n L_\theta^2 \sum_{q_l=0}^{Q_l-1} \mathbb{E}M_{q_l} \left(\sum_{i=0}^{\infty}\rho^i + \frac{2\sum_{i=0}^{\infty}\sqrt{\rho}^i}{1-\sqrt{\rho}}\right)$$

$$\leq \frac{2nQ_l\gamma^2\sigma_l^2}{1-\rho} + \frac{18\gamma^2 n\sigma_g^2 Q_l}{(1-\sqrt{\rho})^2}$$

$$+ \frac{18\gamma^2}{(1-\sqrt{\rho})^2} \sum_{q_l=0}^{Q_l-1} \mathbb{E}\|\nabla f_T(\frac{\Theta_{T,q_l}\mathbf{1}_n}{n}) \cdot \mathbf{1}_n^T\|^2$$

$$+ \frac{18\gamma^2 n L_\theta^2}{(1-\sqrt{\rho})^2} \sum_{q_l=0}^{Q_l-1} \mathbb{E}M_{q_l}$$

Rearranging the terms, we can get:

$$(1 - \frac{18\gamma^2 nL^2}{(1-\sqrt{\rho})^2}) \sum_{q_l=0}^{Q_l-1} \mathbb{E}M_{q_l} \leq \frac{2nQ_l\gamma^2\sigma_l^2}{1-\rho} + \frac{18\gamma^2 n\sigma_g^2 Q_l}{(1-\sqrt{\rho})^2}$$

$$+ \frac{18\gamma^2}{(1-\sqrt{\rho})^2} \sum_{q_l=0}^{Q_l-1} \mathbb{E}\|\nabla f_T(\frac{\Theta_{T,q_l}\mathbf{1}_n}{n}) \cdot \mathbf{1}_n^T\|^2$$

Denote $\Phi_1 = (1 - \frac{18\gamma^2 nL^2}{(1-\sqrt{\rho})^2})$, we can get:

$$\sum_{q_l=0}^{Q_l-1} \mathbb{E}M_{q_l} \leq \frac{2nQ_l\gamma^2\sigma_l^2}{(1-\rho)\Phi_1}$$

$$+ \frac{18\gamma^2 n\sigma_g^2 Q_l}{(1-\sqrt{\rho})^2\Phi_1}$$

$$+ \frac{18\gamma^2}{(1-\sqrt{\rho})^2\Phi_1} \sum_{q_l=0}^{Q_l-1} \mathbb{E}\|\nabla f_T(\frac{\Theta_{T,q_l}\mathbf{1}_n}{n}) \cdot \mathbf{1}_n^T\|^2$$

Return to $T_{21}$:

$$T_{21} \leq \frac{L_\theta^2}{n} \sum_{i=1}^{n} R_q^{(i)} = L^2\mathbb{E}M_{q_l}$$

To bound $T_3$ (Using $\|a+b\|^2 - \|a\|^2 - \|b\|^2 = 2\langle a, b\rangle$):

$$T_3 = -\gamma\mathbb{E}\left\langle \nabla f_{1:T-1}\left(\frac{\Theta_{T,q_l}\mathbf{1}_n}{n}\right), \nabla f_T\left(\frac{\Theta_{T,q_l}\mathbf{1}_n}{n}\right)\right\rangle$$

$$= -\frac{\gamma}{2}\mathbb{E}\left(\|\nabla f_{1:T}\left(\frac{\Theta_{T,q_l}\mathbf{1}_n}{n}\right)\|^2\right.$$

$$\left. - \|\nabla f_{1:T-1}\left(\frac{\Theta_{T,q_l}\mathbf{1}_n}{n}\right)\|^2 - \|\nabla f_T\left(\frac{\Theta_{T,q_l}\mathbf{1}_n}{n}\right)\|^2\right)$$

$$= -\frac{\gamma}{2}\mathbb{E}\|\nabla f_{1:T}\left(\frac{\Theta_{T,q_l}\mathbf{1}_n}{n}\right)\|^2$$

$$+ \frac{\gamma}{2}\mathbb{E}\|\nabla f_{1:T-1}\left(\frac{\Theta_{T,q_l}\mathbf{1}_n}{n}\right)\|^2 + \frac{\gamma}{2}\mathbb{E}\|\nabla f_T\left(\frac{\Theta_{T,q_l}\mathbf{1}_n}{n}\right)\|^2$$

$$= -\frac{\gamma}{2}\mathbb{E}\|\nabla f_{1:T}\left(\frac{\Theta_{T,q_l}\mathbf{1}_n}{n}\right)\|^2 + \frac{\gamma}{2}\mathbb{E}\|\nabla f_{1:T-1}\left(\frac{\Theta_{T,q_l}\mathbf{1}_n}{n}\right)$$

$$- \nabla f_{1:T-1}\left(\frac{\Theta_{T,0}\mathbf{1}_n}{n}\right) + \nabla f_{1:T-1}\left(\frac{\Theta_{T,0}\mathbf{1}_n}{n}\right)\|^2$$

$$+ \frac{\gamma}{2}\mathbb{E}\|\nabla f_T\left(\frac{\Theta_{T,q_l}\mathbf{1}_n}{n}\right)\|^2$$

$$= -\frac{\gamma}{2}\mathbb{E}\|\nabla f_{1:T}\left(\frac{\Theta_{T,q_l}\mathbf{1}_n}{n}\right)\|^2 + \frac{\gamma^3 L^2(T-1)^2}{2}\frac{\sigma_l^2}{n}$$

$$+ \frac{\gamma}{2}\mathbb{E}\|\nabla f_{1:T-1}\left(\frac{\Theta_{T,0}\mathbf{1}_n}{n}\right)\|^2 + \frac{\gamma}{2}\mathbb{E}\|\nabla f_T\left(\frac{\Theta_{T,q_l}\mathbf{1}_n}{n}\right)\|^2$$

Plugging $T_{21}$ into $T_2$, we can get:

$$T_2 \leq \frac{\gamma T}{2}\mathbb{E}\|\nabla f_T\left(\frac{\Theta_{T,q_l}\mathbf{1}_n}{n}\right)\|^2 + \frac{\gamma(T+1)}{2}L^2\mathbb{E}M_{q_l} + \frac{\gamma T^2}{2}\sigma_t^2$$

Plugging $T_2, T_3$ into $T_1$ we can get:

$$T_1 \leq T_2 + T_3 - \gamma\mathbb{E}\|\nabla f_T\left(\frac{\Theta_{T,q_l}\mathbf{1}_n}{n}\right)\|^2$$

$$\leq \frac{\gamma T}{2}\mathbb{E}\|\nabla f_T\left(\frac{\Theta_{T,q_l}\mathbf{1}_n}{n}\right)\|^2 + \frac{\gamma(T+1)}{2}L^2\mathbb{E}M_{q_l} + \frac{\gamma T^2}{2}\sigma_t^2 - \gamma\mathbb{E}\|\nabla f_T\left(\frac{\Theta_{T,q_l}\mathbf{1}_n}{n}\right)\|^2$$

$$+ \frac{\gamma^3 L^2(T-1)^2}{2}\frac{\sigma_l^2}{n} + \frac{\gamma}{2}\mathbb{E}\|\nabla f_{1:T-1}\left(\frac{\Theta_{T,0}\mathbf{1}_n}{n}\right)\|^2 - \frac{\gamma}{2}\mathbb{E}\|\nabla f_{1:T}\left(\frac{\Theta_{T,q_l}\mathbf{1}_n}{n}\right)\|^2 + \frac{\gamma}{2}\mathbb{E}\|\nabla f_T\left(\frac{\Theta_{T,q_l}\mathbf{1}_n}{n}\right)\|^2$$

$$\leq -\frac{\gamma}{2}\mathbb{E}\|\nabla f_{1:T}\left(\frac{\Theta_{T,q_l}\mathbf{1}_n}{n}\right)\|^2 + \frac{\gamma^3 L^2(T-1)^2}{2}\frac{\sigma_l^2}{n} + \frac{\gamma}{2}\mathbb{E}\|\nabla f_{1:T-1}\left(\frac{\Theta_{T,0}\mathbf{1}_n}{n}\right)\|^2$$

$$+ \frac{\gamma(T-1)}{2}\mathbb{E}\|\nabla f_T\left(\frac{\Theta_{T,q_l}\mathbf{1}_n}{n}\right)\|^2 + \frac{\gamma(T+1)}{2}L^2\mathbb{E}M_{q_l} + \frac{\gamma T^2}{2}\sigma_t^2$$

Thus we can get:

$$\mathbb{E}f_{1:T}\left(\frac{\Theta_{T,q_l+1}\mathbf{1}_n}{n}\right)$$

$$\leq\mathbb{E}f_{1:T}\left(\frac{\Theta_{T,q_l}\mathbf{1}_n}{n}\right) + T_1 + \frac{\gamma^2 L}{2n}\sigma_l^2 + \frac{\gamma^2 L}{2}\mathbb{E}\|\frac{\partial f_T(\Theta_{T,q_l})\mathbf{1}_n}{n}\|^2$$

$$\leq\mathbb{E}f_{1:T}\left(\frac{\Theta_{T,q_l}\mathbf{1}_n}{n}\right) - \frac{\gamma}{2}\mathbb{E}\|\nabla f_{1:T}\left(\frac{\Theta_{T,q_l}\mathbf{1}_n}{n}\right)\|^2 + \frac{\gamma}{2}\mathbb{E}\|\nabla f_{1:T-1}\left(\frac{\Theta_{T,0}\mathbf{1}_n}{n}\right)\|^2 + \frac{\gamma(T-1)}{2}\mathbb{E}\|\nabla f_T\left(\frac{\Theta_{T,q_l}\mathbf{1}_n}{n}\right)\|^2$$

$$+ \frac{\gamma^2 L}{2}\mathbb{E}\|\frac{\partial f_T(\Theta_{T,q_l})\mathbf{1}_n}{n}\|^2 + \frac{\gamma(T+1)L^2}{2}\mathbb{E}M_{q_l} + \frac{\gamma^2 L}{2n}(1+\gamma L(T-1)^2)\sigma_l^2 + \frac{\gamma T^2}{2}\sigma_t^2$$

Summing from $q_l = 0$ to $q_l = Q_l$, we get:

$$\frac{\gamma}{2}\sum_{q_l=0}^{Q_l-1}\mathbb{E}\|\nabla f_{1:T}\left(\frac{\Theta_{T,q_l}\mathbf{1}_n}{n}\right)\|^2$$

$$\leq \underbrace{\mathbb{E}f_{1:T}\left(\frac{\Theta_{T,0}\mathbf{1}_n}{n}\right) - \mathbb{E}f_{1:T}\left(\frac{\Theta_{T,Q_l}\mathbf{1}_n}{n}\right)}_{\Delta_f} + \frac{\gamma Q_l}{2}\mathbb{E}\|\nabla f_{1:T-1}\left(\frac{\Theta_{T,0}\mathbf{1}_n}{n}\right)\|^2 + \frac{\gamma(T-1)}{2}\sum_{q_l=0}^{Q_l-1}\mathbb{E}\|\nabla f_T\left(\frac{\Theta_{T,q_l}\mathbf{1}_n}{n}\right)\|^2$$

$$+ \frac{\gamma^2 L}{2}\sum_{q_l=0}^{Q_l-1}\mathbb{E}\|\frac{\partial f_T(\Theta_{T,q_l})\mathbf{1}_n}{n}\|^2 + Q_l\left(\frac{\gamma^2 L}{2n}(1+\gamma L(T-1)^2)\sigma_l^2 + \frac{\gamma T^2}{2}\sigma_t^2\right) + \frac{\gamma(T+1)L^2}{2}\sum_{q_l=0}^{Q_l-1}\mathbb{E}M_{q_l}$$

Plugging the bound of $\sum_{q_l=0}^{Q_l-1}\mathbb{E}M_{q_l}$:

$$\frac{\gamma(T+1)L^2}{2}\sum_{q_l=0}^{Q_l-1}\mathbb{E}M_{q_l}$$

$$\leq \frac{\gamma(T+1)L^2}{2}\left[\frac{2nQ_l\gamma^2\sigma_l^2}{(1-\rho)\Phi_1} + \frac{18\gamma^2 n\sigma_g^2 Q_l}{(1-\sqrt{\rho})^2\Phi_1} + \frac{18\gamma^2 n}{(1-\sqrt{\rho})^2\Phi_1}\sum_{q_l=0}^{Q_l-1}\mathbb{E}\|\nabla f_T\left(\frac{\Theta_{T,q_l}\mathbf{1}_n}{n}\right)\|^2\right]$$

$$= Q_l\left[\frac{\gamma^3(T+1)L^2 n}{(1-\rho)\Phi_1}\sigma_l^2 + \frac{9\gamma^3(T+1)L^2 n}{(1-\sqrt{\rho})^2\Phi_1}\sigma_g^2\right] + \frac{9\gamma^3(T+1)L^2 n}{(1-\sqrt{\rho})^2\Phi_1}\sum_{q_l=0}^{Q_l-1}\mathbb{E}\|\nabla f_T\left(\frac{\Theta_{T,q_l}\mathbf{1}_n}{n}\right)\|^2$$

Finally, we can get:

$$\frac{\gamma}{2}\sum_{q_l=0}^{Q_l-1}\mathbb{E}\|\nabla f_{1:T}\left(\frac{\Theta_{T,q_l}\mathbf{1}n}{n}\right)\|^2$$

$$\leq \Delta_f + \frac{\gamma Q_l}{2}\mathbb{E}\|\nabla f_{1:T-1}\left(\frac{\Theta_{T,0}\mathbf{1}n}{n}\right)\|^2 + \left(\frac{\gamma(T-1)}{2} + \frac{9\gamma^3(T+1)L^2 n}{(1-\sqrt{\rho})^2\Phi_1}\right)\sum_{q_l=0}^{Q_l-1}\mathbb{E}\|\nabla f_T\left(\frac{\Theta_{T,q_l}\mathbf{1}n}{n}\right)\|^2$$

$$+ \frac{\gamma^2 L}{2}\sum_{q_l=0}^{Q_l-1}\mathbb{E}\|\frac{\partial f_T(\Theta_{T,q_l})\mathbf{1}_n}{n}\|^2 + Q_l\left(\frac{\gamma^2 L}{2n}(1+\gamma L(T-1)^2)\sigma_l^2 + \frac{\gamma T^2}{2}\sigma_t^2 + \frac{\gamma^3(T+1)L^2 n}{(1-\rho)\Phi_1}\sigma_l^2 + \frac{9\gamma^3(T+1)L^2 n}{(1-\sqrt{\rho})^2\Phi_1}\sigma_g^2\right)$$

Dividing both sides of the inequality by $\frac{\gamma Q_l}{2}$:

$$\frac{1}{Q_l}\sum_{q_l=0}^{Q_l-1}\mathbb{E}\|\nabla f_{1:T}\left(\frac{\Theta_{T,q_l}\mathbf{1}n}{n}\right)\|^2$$

$$\leq \frac{2\Delta_f}{\gamma Q_l} + \mathbb{E}\|\nabla f_{1:T-1}\left(\frac{\Theta_{T,0}\mathbf{1}_n}{n}\right)\|^2 + \left((T-1) + \frac{18\gamma^2(T+1)L^2 n}{(1-\sqrt{\rho})^2\Phi_1}\right)\frac{1}{Q_l}\sum_{q_l=0}^{Q_l-1}\mathbb{E}\|\nabla f_T\left(\frac{\Theta_{T,q_l}\mathbf{1}n}{n}\right)\|^2$$

$$+ \gamma L\frac{1}{Q_l}\sum_{q_l=0}^{Q_l-1}\mathbb{E}\|\frac{\partial f_T(\Theta_{T,q_l})\mathbf{1}_n}{n}\|^2 + \frac{\gamma L}{n}(1+\gamma L(T-1)^2)\sigma_l^2 + T^2\sigma_t^2 + \frac{2\gamma^2(T+1)L^2 n}{(1-\rho)\Phi_1}\sigma_l^2 + \frac{18\gamma^2(T+1)L^2 n}{(1-\sqrt{\rho})^2\Phi_1}\sigma_g^2$$

Let $\gamma = \sqrt{\frac{n}{Q_l}}$, then if the total number of iterate $Q_l \geq \frac{36L^2 n^2}{(1-\sqrt{\rho})^2}$ is sufficiently large, we can get the following convergence rate:

$$\frac{1}{Q_l}\sum_{q_l=0}^{Q_l-1}\mathbb{E}\|\nabla f_{1:T}\left(\frac{\Theta_{T,q_l}\mathbf{1}n}{n}\right)\|^2$$

$$\leq \frac{2\Delta_f}{\sqrt{nQ_l}} + \frac{L}{\sqrt{nQ_l}}\sum_{q_l=0}^{Q_l-1}\mathbb{E}\|\frac{\partial f_T(\Theta_{T,q_l})\mathbf{1}_n}{n}\|^2 + \left((T-1) + \frac{36n^2 L^2(T+1)}{Q_l(1-\sqrt{\rho})^2}\right)\frac{1}{Q_l}\sum_{q_l=0}^{Q_l-1}\mathbb{E}\|\nabla f_T\left(\frac{\Theta_{T,q_l}\mathbf{1}n}{n}\right)\|^2$$

$$+ \mathbb{E}\|\nabla f_{1:T-1}\left(\frac{\Theta_{T,0}\mathbf{1}_n}{n}\right)\|^2 + (\frac{L}{\sqrt{nQ_l}} + \frac{L^2(T-1)^2}{Q_l})\sigma_l^2 + T^2\sigma_t^2 + \frac{1}{Q_l}\left(\frac{4n^2 L^2(T+1)\sigma_l^2}{1-\rho} + \frac{36n^2 L^2(T+1)\sigma_g^2}{(1-\sqrt{\rho})^2}\right)$$

Let $\mathbb{E}\|\nabla f_{1:T-1}\left(\frac{\Theta_{T,0}\mathbf{1}_n}{n}\right)\|^2 \leq C$, where $C$ is a constant, $T^2\sigma_t^2$ is also a constant corresponding to the task. We can gain the convergence rate is:

$$\frac{1}{Q_l}\sum_{q_l=0}^{Q_l-1}\mathbb{E}\|\nabla f_{1:T}\left(\frac{\Theta_{T,q_l}\mathbf{1}_n}{n}\right)\|^2 \leq \mathcal{O}(\frac{1}{\sqrt{nQ_l}} + \frac{1}{Q_l})$$

## F. Unlearning analysis

**Theorem F.1.** *When remove class $c_f$ from the data, let $\mathcal{D}_{c_f}$ be the unlearning dataset, and $\mathcal{D}_{\mathcal{C}\backslash c_f}$ be the remaining dataset. There exists $\beta_{ul} < \frac{\epsilon}{2\Gamma}$, we can get the following requirements are satisfied.*

$$\begin{cases} \textbf{R1:} \arg\max_b f^b_{\Theta^l_{new}}(x) = \arg\max_b f^b_{\Theta^l_{ori}}(x), & x \in \mathcal{D}_{\mathcal{C}\backslash c_f}, \\ \textbf{R2:} \arg\max_b f^b_{\Theta^l_{new}}(x) \neq c_f, & x \in \mathcal{D}_{c_f}, \end{cases} \tag{21}$$

*where $\Theta^l_{new} = \frac{1}{n}\sum_{i=1}^n \theta^l_{new}$ and $\Theta^l_{ori} = \frac{1}{n}\sum_{i=1}^n \theta^l_{ori}$. $\Gamma$ is the upper bound of the logits generated by $\theta^l_{ul}$ on remain distribution, that is $\Gamma = \max_{x\in\mathcal{D}_{\mathcal{C}\backslash c_f}, b\in[\mathcal{C}|\mathcal{T}|]} f^b_{\theta^l_{ul}}(x)$ and $\epsilon$ is the margin between the largest and the second largest value of $f^b_{\Theta^l_{ori}}(x)$.*

Here we denote the unlearning method should satisfy the following two requirements:

$$\begin{cases} \textbf{R1:} \arg\max_b f^b_{\Theta^l_{new}}(x) = \arg\max_b f^b_{\Theta^l_{ori}}(x), & x \in \mathcal{D}_{\mathcal{C}\backslash c_f}, \\ \textbf{R2:} \arg\max_b f^b_{\Theta^l_{new}}(x) \neq c_f, & x \in \mathcal{D}_{c_f}, \end{cases} \tag{22}$$

where $\Theta^l_{new} = \frac{1}{n} \sum_{i=1}^{n} \theta^l_{new}$ and $\Theta^l_{ori} = \frac{1}{n} \sum_{i=1}^{n} \theta^l_{ori}$.

As Eq.9 shows, $\theta^l_{new} := \theta^l_{ori} - \beta_{ul}\theta^l_{ul}$. Since $\theta^l_{ul}$ is learned by $\hat{\mathcal{D}}_f$, whose classes are all label $c_f$, we have

$$\arg\max_b f^b_{\theta^l_{ul}}(x) = c_f, \quad x \in \mathcal{D}_{\mathcal{C} \setminus c_f}.$$

For the requirement R1: For $x \in \mathcal{D}_{\mathcal{C} \setminus c_f}$, let $\epsilon$ be the margin between the largest and the second largest value of $f^b_{\Theta^l_{ori}}(x)$. If the unlearning module's response on this sample is bounded by $\Gamma = \max_{x \in \mathcal{D}_{\mathcal{C} \setminus c_f}, b \in [\mathcal{C}^{|\mathcal{T}|}]} f^b_{\theta^l_{ul}}(x)$, then the prediction remains unchanged, that is,

$$\arg\max_b f^b_{\Theta^l_{new}}(x) \tag{23}$$

$$= \arg\max_b f^b_{\frac{1}{n}\sum_{i=1}^{n}(\theta^l_{ori} - \beta_{ul}\theta^l_{ul})}(x) \tag{24}$$

$$= \arg\max_b f^b_{\Theta^l_{ori} - \beta_{ul}\Theta^l_{ul}}(x) \tag{25}$$

$$= \arg\max_b \left( f^b_{\Theta^l_{ori}}(x) - \beta_{ul} f^b_{\Theta^l_{ul}}(x) \right) \tag{26}$$

$$= \arg\max_b f^b_{\Theta^l_{ori}}(x), \tag{27}$$

where the third equality is due to the linear nature of the fully connected layer $\theta^l$ and the forth equality is established when $\beta_{ul} < \frac{\epsilon}{2\Gamma}$.

For the requirement R2: For any target sample $(x, c_f) \in \mathcal{D}_{c_f}$, there exists a coefficient $\beta_{ul} > 0$ such that we can obtain:

$$\arg\max_b f^b_{\Theta^l_{new}}(x) \tag{28}$$

$$= \arg\max_b f^b_{\Theta^l_{ori} - \beta_{ul}\Theta^l_{ul}}(x) \tag{29}$$

$$= \arg\max_b \left( f^b_{\Theta^l_{ori}}(x) - \beta_{ul} f^b_{\Theta^l_{ul}}(x) \right) \tag{30}$$

$$\neq c_f, \tag{31}$$

This inequality holds because the weighted subtraction reduces the value of class $c_f$ below that of the remaining classes.

