# OpenReview forum: "Forgetting Whenever You Want: A Decentralized Continual Learning Framework with On-Demand Unlearning"
_ICML.cc/2026/Conference — ICML 2026 regular_

### Official Review · Reviewer_PFT6 · 2026-02-24

**Soundness:** 3
**Presentation:** 3
**Significance:** 3
**Originality:** 4
**Overall Recommendation:** 4
**Confidence:** 4

**Summary:**

This study investigates decentralized continual learning with on demand unlearning under privacy constraints prohibiting raw data storage. The authors propose DCU, which leverages frozen diffusion models to extract class specific prototypes for synthetic replay and distillation. Unlearning is achieved through boundary adjustment using synthetic data and classification head subtraction. Experiments on RGB and Grayscale benchmarks under heterogeneous settings demonstrate improved retention and near zero target accuracy, validating the utility of disposable prototypes for knowledge retention and selective forgetting.

**Compliance With Llm Reviewing Policy:**

Affirmed.

**Key Questions For Authors:**

1. How do you verify that unlearning removes target class information beyond top one accuracy degradation on target classes, especially given that the feature extractor is frozen during unlearning and only the classification head is modified through fine tuning and subtraction？
2. Can you clarify which parts of the theoretical convergence result apply directly to the full DCU training objective and task stream, and which parts apply only to the decentralized optimization subroutine under fixed objectives？
3. Have you tested whether unlearning remains effective under repeated unlearning requests on semantically similar classes or under class overlap settings beyond strictly disjoint task class partitions？

**Limitations:**

The study is limited to benchmark scale image datasets and controlled class incremental splits, so it remains unclear how well the approach scales to larger models, richer modalities, or more realistic decentralized environments with stronger heterogeneity and communication constraints.

**Strengths And Weaknesses:**

Strengths：The problem formulation is novel and significant. This work addresses the underexplored intersection of decentralized learning, continual learning, and class unlearning without historical raw data. The motivation stems from privacy constraints and right to be forgotten requirements. The method design is coherent and modular. A three stage structure connects prototype extraction, guided continual learning, and unlearning consistently. The same prototype mechanism supports both replay and forgetting. The empirical section includes multiple baselines, datasets, and non IID settings. An ablation on synthetic data components supports the claimed contributions.

Weakness：The unlearning evaluation relies on accuracy proxies rather than evidence of true information removal from representations. Formal requirements are stated but experiments focus on remaining and target class accuracy. It remains unclear whether residual information persists in the frozen feature extractor. Forgetting may be limited to logit level suppression. The theoretical guarantee appears disconnected from the training pipeline. The convergence statement applies to a specific subroutine under specific conditions. The practical objective involves many components like synthetic data and weighted cross entropy. It is unclear whether the bound characterizes the end to end dynamics. The empirical protocol raises fairness concerns. Experiments use small benchmarks with limited classes. Implementation details for baseline adaptations are deferred to the appendix. Results emphasize accuracy without accounting for computational cost or communication overhead.

---

> ### Author Rebuttal · Authors · 2026-03-31
>
> Dear reviewer, we sincerely appreciate your valuable comments and constructive suggestions.
>
> **Response to W1:** Our unlearning approach primarily follows FedAU, which defines unlearning in a generalized sense as the suppression of knowledge output rather than the destruction of the underlying feature representation.
>
> **Response to W2:** Theorem E.6 characterizes the dynamics of the full DCU training objective across the entire task stream.
> We treat the combined loss $\mathcal{L}_{cl}$ as the optimization target for each task and mathematically capture the evolution of the task stream by Assumption E.5. It bounds the gradient differences between sequential tasks. Since prototypes are extracted from a frozen pre-trained model and remain constant after aggregation, we treat them as fixed environmental parameters within the convergence analysis.
> Otherwise, the unlearning process is a discrete operation validated separately by the logit margin requirements in Theorem F.1.
>
> **Response to W3:** We conducted continuous unlearning immediately after learning the first 10 classes on the RGB dataset. Notably, this class set includes bear and beaver, which are both brown animals sharing certain semantic similarities.
>
> | | $cl_{0:9}$|$ul_{baby}$|$ul_{apple}$|$ul_{fish}$|$ul_{bear}$|$ul_{beaver}$|
> |--|--|--|--|--|--|--|
> |Ours|75.14| 73.27| 71.73| 71.69| 74.13| 70.96|
>
> The final per-class top-1 accuracy distribution is ['0.00', '0.00', '0.00', '0.00', '0.00', '90.00', '69.40', '66.00', '88.00', '41.40'], demonstrating excellent unlearning performance across the targeted classes.
>
> Thank you again for the valuable comments. We hope you might find the response satisfactory.

---

> > ### Author Rebuttal · Reviewer_PFT6 · 2026-04-02
> >
> > Thank you for the detailed rebuttal. I appreciate the clarification that your notion of unlearning follows the generalized suppression based definition adopted in prior work, the additional explanation regarding the scope of Theorem E.6 and the separate role of Theorem F.1, as well as the new experiment on repeated unlearning over semantically related classes. That said, my overall assessment and score remain unchanged. The response to the first concern confirms that the method mainly targets output level suppression rather than removal of class information from internal representations, so the original concern about the strength of the unlearning claim still stands. For the second point, although you state that Theorem E.6 covers the full task stream, the current explanation still leaves some ambiguity about how tightly the assumptions match the practical DCU pipeline with synthetic replay, prototype aggregation, and weighted objectives, while the unlearning step is analyzed separately rather than in a unified end to end manner. For the third point, the added results are useful, but they remain limited in scope and do not fully address more challenging overlap or broader semantic similarity scenarios.

---

> > > ### Author Response · Authors · 2026-04-03
> > >
> > > We sincerely thank you for the thoughtful feedback and for acknowledging the clarifications provided in our rebuttal regarding the definition of unlearning, the scope of Theorem E.6, and the additional experiments on semantically related classes. We will delve deeper into the underlying mechanisms of unlearning and strengthen our theoretical considerations in future research. We appreciate the time taken to re-evaluate our work.

---

### Official Review · Reviewer_45PS · 2026-03-11

**Soundness:** 2
**Presentation:** 3
**Significance:** 2
**Originality:** 2
**Overall Recommendation:** 3
**Confidence:** 4

**Summary:**

This paper proposes DCU, a framework that simultaneously supports incremental continual learning and on demand class forgetting in a decentralized environment. The core innovation lies in introducing class prototypes extracted from a pretrained diffusion model. These prototypes serve as lightweight embeddings of class concepts and can be used to generate synthetic data without storing raw data, thereby supporting 1) continual learning to alleviate catastrophic forgetting, and 2) precise removal of knowledge related to specific classes. Experiments on two datasets verify the effectiveness of the method in dynamic learning and targeted forgetting scenarios, and the paper also provides convergence analysis.

**Compliance With Llm Reviewing Policy:**

Affirmed.

**Final Justification:**

This paper has certain contributions and significance, but there are still some shortcomings. The authors’ response did not resolve all of my concerns. Therefore, I have decided to keep my score unchanged.

**Key Questions For Authors:**

1. The framework heavily relies on all clients sharing a high-quality, pre-trained diffusion model. In practice, if some clients lack access to such a model, or if the model's pre-trained domain does not match the task data, how robust is the prototype extraction? Are there any fallback plans or mitigation strategies?
2. The linear subtraction of parameters (Eq. 10) is central to the unlearning process. Can you provide more intuitive explanations or visualizations (e.g., t-SNE plots of the feature space before and after unlearning, or changes in the classification layer's weights) to demonstrate that this subtraction genuinely "erases" the target class knowledge rather than just causing misclassification? How do you ensure it doesn't negatively impact other, potentially highly coupled, classes?
3. How does DCU behave when scaling up to more clients (e.g., 50 or 100) or larger datasets (full CIFAR-100 or Tiny ImageNet)? Do you observe stability or efficiency issues in such settings?
4. The prototype extraction phase involves multiple communication rounds and local iterations on each client to optimize the prototypes. Compared to standard decentralized continual learning, what is the additional computational and communication overhead introduced by DCU? Could this overhead become a bottleneck, especially when dealing with a large number of classes?
5. Have you considered or measured potential privacy risks from the diffusion model or synthetic data (e.g., memorization or inversion attacks), beyond the fact that raw data are not stored?

**Limitations:**

yes

**Strengths And Weaknesses:**

Strengths:
1. The paper addresses a highly relevant and timely problem by combining continual learning and on-demand unlearning in a decentralized setting for the first time. The problem formulation is clear and has strong practical implications, especially in light of privacy regulations like GDPR.
2. The idea of using pretrained diffusion models to extract "disposable prototypes" is clever and elegant. It effectively addresses the core dilemma in decentralized settings: the need for historical data for replay (in continual learning) and unlearning, while being unable to store raw data due to privacy.
3. The experimental setup is described in reasonable detail, especially in the appendix, including client numbers, communication rounds, local epochs, synthetic data counts, and implementation details, which makes the empirical evaluation relatively reproducible and transparent. The method is systematically compared against combinations of continual learning and unlearning baselines adapted to decentralized settings, and results consistently show improved retention of non-forgotten classes while achieving strong forgetting of target classes.

Weaknesses:
1. The framework heavily relies on a high-quality, pretrained diffusion model shared by all clients, which is a strong and often unrealistic assumption in decentralized settings, especially for domains without powerful pretrained models, and the paper does not analyze how degraded diffusion quality affects performance.
2. The unlearning mechanism based on subtracting classifier parameters is justified under idealized linear and small-step assumptions, which are questionable for deep nonlinear networks, and the paper lacks finer-grained empirical analysis beyond aggregate metrics to validate the precision and robustness of this mechanism.
3. Experiments are limited to a composite MNIST-style grayscale dataset and the first 30 classes of CIFAR-100 with only 5–10 clients, without stress tests on larger datasets or more clients, so scalability and robustness in realistic large-scale decentralized settings remain unproven.
4. There are repeated descriptions and minor formatting issues in the main text, and the lengthy appendix derivations use somewhat inconsistent notation and large logical jumps, making the proofs hard to follow and weakening the perceived rigor.

---

> ### Author Rebuttal · Authors · 2026-03-31
>
> Dear reviewer, we sincerely appreciate your valuable comments and constructive suggestions.
>
> **Response to W1:** Our scenario assumes that users possess a pretrained diffusion model, establishing a setting analogous to large-small model collaboration. In this framework, the pretrained model serves as a foundation for knowledge extraction, thereby supporting both continual learning and targeted knowledge unlearning. Then, we conduct experiments by reducing the number of local iterations in the prototype extraction phase from the default 50 to 40 and 30. As shown in the table, even when the iteration steps are reduced to 60% of the original (leading to undertrained prototypes), the final performance is not significantly affected.
> | | $cl_{0:9}$|$cl_{10:19}$|$ul_{2}$|$cl_{20:29}$|$ul_{20}$|
> |--|--|--|--|--|--|
> |30|78.26| 55.74| 55.78| 50.68| 49.98|
> |40|77.22| 59.89| 58.61| 52.83| 51.52|
> |50(default)|76.56| 58.85| 58.14| 51.14| 50.06|
>
> **Response to W2:** Our ul method primarily focuses on class-wise unlearning at the output level; therefore, it only necessitates processing the linear layer. The unlearning process is a discrete operation validated separately by the logit margin requirements in Theorem F.1.
>
>
> **Response to W3:** We additionally conduct partial experiments on the first 30 classes of ImageNet, following the same setting used for the RGB dataset.
>
> | | $cl_{0:9}$|$cl_{10:19}$|$ul_{2}$|$cl_{20:29}$|$ul_{20}$|
> |--|--|--|--|--|--|
> |lwf_au|44.4| 32.4|--| 23.87| 20.62|
> |target_au|44.4| 36.0| 35.47| 25.93| 26.29|
> |clmul| 44.4| 31.1| 21.79| 27.86| 24.43|
> |Ours | 51.24| 37.88| 36.46| 27.86| 27.47|
>
>
> **Response to Q4:** In terms of storage, each class prototype is a lightweight 1280-dimensional vector, occupying only 0.005 MB. Regarding time cost, we primarily report training-related per-client timings on the RGB dataset. Specifically, the prototype training time, model training time, and unlearning time, as summarized in the table. The data generation for each class may incur considerable overhead due to our use of a naive DDIM sampling strategy. However, this can be mitigated through inference acceleration techniques.
>
> ||prototype training|learning|unlearning|data generation for each class
> |--|--|--|--|--|
> |time cost(min)|$\approx$ 5| $\approx$ 14.5|$\approx$ 0.15|$\approx$ 3.6
> |rounds|10|50|-|-|
>
> **Response to W5&Q5:** There is a substantial body of widely recognized work [1][2] that enhances federated learning by uploading class prototypes or synthetic data. We argue that class prototypes offer stronger privacy guarantees compared to transmitting raw data. While sophisticated attacks could potentially pose risks, such considerations fall outside the scope of this work.
>
> [1] Fedproto: Federated prototype learning across heterogeneous clients
> [2] Tackling data heterogeneity in federated learning with class prototypes
> [3] DaFKD: Domain-aware Federated Knowledge Distillation
>
> Thank you again for the valuable comments. We hope you might find the response satisfactory.

---

> > ### Author Rebuttal · Reviewer_45PS · 2026-04-03
> >
> > Thank you to the authors for the detailed response and the additional experiments. The response has alleviated some of my concerns to a certain extent, especially through the added sensitivity study on the number of prototype extraction steps, the additional partial ImageNet results, and the clarification of some storage and time overheads.
> >
> > However, in my view, the authors have not fully resolved my concerns. First, regarding the method’s reliance on a high-quality pretrained diffusion model, both the main paper and the method design itself take such a model as a prerequisite. The newly added experiments in the response mainly show that performance does not degrade significantly when the number of prototype extraction iterations is reduced, but this does not truly answer the question of whether the method remains robust when the pretraining domain does not match the task domain. The authors only added experiments on the first 30 classes of ImageNet, rather than providing full validation on CIFAR-100 or Tiny-ImageNet. In addition, the computation overhead comparison still lacks comparison with standard decentralized continual learning methods.
> >
> > Second, regarding the parameter-subtraction-based unlearning in Eq. (10), the authors do provide a clear mechanism in the paper: they freeze the feature extractor, perform unlearning only on the output layer, and realize forgetting through subtraction at the classification head. However, the empirical evidence is still mainly based on aggregate metrics, and still lacks the more fine-grained analyses I mentioned earlier, such as changes in the feature space, the evolution of classifier weights, or visualization and verification of the effects on highly coupled classes.
> >
> > Moreover, regarding the potential risks of memorization, inversion, or membership inference attacks associated with the diffusion model or synthetic data, the authors still largely treat these concerns as being beyond the scope of this paper. Therefore, this part of my concern has not been genuinely addressed either. As a result, I am more inclined to maintain my original judgment and keep my current score.

---

> > > ### Author Response · Authors · 2026-04-05
> > >
> > > We sincerely thank the reviewer for the constructive feedback. We agree that the points raised are important; however, our work primarily focuses on leveraging existing pretrained models to bridge Unlearning and Continual Learning. We will take the reviewer's suggestions into consideration and explore pretrained diffusion models more thoroughly in our future research. Thank you once again for the valuable comments.

---

### Official Review · Reviewer_3x8s · 2026-03-12

**Soundness:** 3
**Presentation:** 3
**Significance:** 3
**Originality:** 3
**Overall Recommendation:** 4
**Confidence:** 5

**Summary:**

This paper proposes a novel decentralized continual learning framework with on-demand unlearning (DCU). Addressing the two major challenges of "Historical Class Unlearning" and "Network-Wide Knowledge Entanglement" under the constraints of no centralized server and irreversible unavailability of historical data, the authors ingeniously introduce pre-trained diffusion models to extract lightweight and disposable class prototypes. During the continual learning phase, these prototypes are utilized to generate synthetic data to mitigate catastrophic forgetting and combat non-IID data distributions. In the unlearning phase, historical data distributions are reconstructed via prototypes to guide the fine-tuning of the decision boundary, after which the corresponding prototypes are discarded to preserve privacy.

**Compliance With Llm Reviewing Policy:**

Affirmed.

**Key Questions For Authors:**

1. The paper mentions that clients utilize frozen pre-trained diffusion models (DMs) for prototype extraction and synthetic data generation. Did the pre-trained DM observe images from CIFAR-100 or other related datasets during its pre-training phase? If so, when the system receives a request to unlearn class $c_f$, even if this class is erased at the classification head level, does the underlying DM still "remember" how to generate images of this class? Could this pose a vulnerability under strict privacy compliance requirements (e.g., the need to completely eradicate a specific concept)?

2. According to Eq. (10), the unlearning operation is primarily achieved via a linear subtraction on the classification head $\theta^l$: $\theta_{new}^{l}:=\theta_{ori}^{l}-\beta_{ul}\theta_{ul}^{l}$.Question: The entire feature extractor $\theta^e$ appears to remain frozen during the unlearning process. Does this imply that while the classifier can no longer output the label for class $c_f$, the feature extractor $\theta^e$ still retains the ability to extract highly discriminative features for class $c_f$? Is this merely a masking effect rather than genuine parameter-level unlearning?

3. The experimental section primarily demonstrates scenarios alternating between learning and single/double unlearning requests. If the system receives high-frequency unlearning requests (e.g., 10 or 20 different classes) over a long period, repeatedly generating noisy data and performing subtraction operations, will the performance on the retained classes suffer from irreversible collapse? Could the authors provide a stress test to demonstrate the resilience and upper limits of the DCU framework?

4. During the prototype extraction phase, clients perform a linear weighted average of the extracted feature prototypes $p_{c,i}$ via the communication matrix $W$: $p_{c,i}=\sum_{j=1}^{n}W_{ij}p_{c,j}$. Under highly non-IID settings (e.g., $Dir(0.1)$), data distributions across some clients are extremely skewed. In the condition embedding space of the DMs, does directly applying a linear average to prototypes from different clients still mathematically represent a meaningful global image manifold? Could this lead to semantic collapse in the generated synthetic data?

5. Need to analyze the communication time complexity and computational time complexity with SOTA.

6. It appears that the formatting for the mathematical components (e.g., Assumptions, Theorems, and Proofs), particularly in the Appendix, does not strictly adhere to the standard ICML formatting guidelines?

**Limitations:**

Yes

**Strengths And Weaknesses:**

Strengths

To the best of my knowledge, this is the first work to explore simultaneous continual learning and machine unlearning under the strict constraints of a "decentralized" environment with "data-free" requirements. This problem formulation is highly practical and aligns perfectly with increasingly stringent data privacy regulations (e.g., GDPR).

The framework effectively utilizes frozen pre-trained diffusion models to extract disposable prototypes. This data-free generative replay mechanism not only resolves raw data storage privacy concerns but also simplifies complex feature disentanglement into a lightweight boundary adjustment operation.

The paper provides clear theoretical backing, including convergence guarantees for D-PSGD updates in a decentralized topology and rigorous proofs demonstrating that the unlearning operations satisfy the criteria for accurately removing specific target classes without harming remaining knowledge.

Weaknesses

DCU heavily depends on local pre-trained DMs for prototype extraction and data generation. If new continual learning tasks involve rare concepts or domain data never seen by the pre-trained DM (Out-of-Distribution), will the generation quality and subsequent unlearning precision of this framework degrade significantly? It is recommended that the authors add discussions or experiments regarding cross-domain or non-natural image tasks to clarify the boundary of the model's capabilities.

The paper assumes clients exchange prototype information with neighboring nodes via a dynamic topology matrix $W$. Although prototypes are lightweight embeddings, detailed quantitative analyses or comparisons regarding the communication costs of frequent prototype aggregation and the computational costs of local DM inference (Time/Communication/Computation Overhead) in large-scale networks or low-bandwidth edge devices are somewhat lacking in the main text.

 As observed in the experimental results, while DCU performs exceptionally well on the Grayscale dataset, the accuracy on remaining learned classes drops noticeably on the more complex RGB dataset (e.g., maintaining only around $50.67\%$ as shown in Table 1). How do the authors view this performance bottleneck? Are there further optimization strategies planned for more complex, high-dimensional feature spaces?

---

> ### Author Rebuttal · Authors · 2026-03-31
>
> Dear reviewer, we sincerely appreciate your valuable comments and constructive suggestions.
>
> **Response to W1&Q1:** On Grayscale, we used a special grayscale diffusion model for small-scale testing, which is difficult to extend to out-of-domain tasks. However, for the RGB experiments, we used Stable Diffusion, a general-purpose model trained on the LAION dataset. Although it has no direct dependency on the CIFAR-100 dataset used in the experiments, it possesses strong generalization capability, which serves as the foundation for our prototype extraction on RGB tasks. We additionally conducted experiments on an ImageNet subset to demonstrate its generalization ability. The diffusion model is only used to extract prototypes and does not memorize knowledge from the generated category embeddings.
>
> | | $cl_{0:9}$|$cl_{10:19}$|$ul_{2}$|$cl_{20:29}$|$ul_{20}$|
> |--|--|--|--|--|--|
> |lwf_au|44.4| 32.4|--| 23.87| 20.62|
> |target_au|44.4| 36.0| 35.47| 25.93| 26.29|
> |clmul| 44.4| 31.1| 21.79| 27.86| 24.43|
> |Ours | 51.24| 37.88| 36.46| 27.86| 27.47|
>
> **Response to W2:** In terms of communication cost, each class prototype is a lightweight 1280-dimensional vector, occupying only 0.005 MB. Since the embedding size is fixed, the storage and communication costs are independent of the number of clients. Regarding time cost, we primarily report training-related per-client timings on the RGB dataset. Specifically, the prototype training time, model training time, and unlearning time, as summarized in the table. The data generation for each class may incur considerable overhead due to our use of a naive DDIM sampling strategy. However, this can be mitigated through inference acceleration techniques.
>
> ||prototype training|learning|unlearning|data generation for each class
> |--|--|--|--|--|
> |time cost(min)|$\approx$ 5| $\approx$ 14.5|$\approx$ 0.15|$\approx$ 3.6
> |rounds|10|50|-|-|
>
>
> **Response to Q2:** Our unlearning approach primarily follows FedAU, which defines unlearning in a generalized sense as the suppression of knowledge output rather than the destruction of the underlying feature representation. Note that unlearning is achieved through linear operations, which is logically different from directly applying a mask to the unlearned categories.
>
> **Response to Q3:** We conducted continuous unlearning immediately after learning the first 10 classes on the RGB dataset.
>
> | | $cl_{0:9}$|$ul_{0}$|$ul_{1}$|$ul_{2}$|$ul_{3}$|$ul_{4}$|
> |--|--|--|--|--|--|--|
> |Ours|75.14| 73.27| 71.73| 71.69| 74.13| 70.96|
>
> The final per-class top-1 accuracy distribution is ['0.00', '0.00', '0.00', '0.00', '0.00', '90.00', '69.40', '66.00', '88.00', '41.40'], demonstrating excellent unlearning performance across the targeted classes.
>
>
> **Response to W3:** Distributed continual learning remains a challenging area that requires further research. Referring to the continual learning results presented in our experimental section, this approach itself has inherent limitations when applied to complex tasks. We focus more on the intersection of achieving unlearning in the context of continual learning.
>
> **Response to Q4&Q6:** We appreciate the feedback. In the revised version, we will include examples of synthetic data under varying degrees of non-IID data distributions and improve the formatting of the formulas.
>
>
> Thank you again for the valuable comments. We hope you might find the response satisfactory.

---

> > ### Author Rebuttal · Reviewer_3x8s · 2026-04-02
> >
> > I would like to thank the authors for their detailed and honest rebuttal. While I appreciate their clarifications, several of my major concerns regarding the fundamental premise and practicality of the proposed DCU framework remain partially unresolved:
> >
> > Superficial Unlearning: As admitted in Response to Q2, the unlearning is primarily "the suppression of knowledge output" rather than true feature destruction. This "masking" at the classification head leaves potential vulnerabilities under strict privacy requirements.
> >
> > Computational Overhead: The reported 3.6 minutes per class generation time using DDIM is still quite heavy for edge devices in a realistic decentralized setting.
> >
> > Reliance on Massive Priors: Utilizing Stable Diffusion implies the model heavily relies on a massive global prior, which somewhat sidesteps the strict "data-free" constraints in sequential learning.
> >
> >
> > Despite the aforementioned structural and practical flaws, I personally find the problem setting—simultaneous continual learning and on-demand unlearning in a decentralized environment—to be extremely interesting, highly relevant, and a much-needed exploration in the community. Because of my strong personal interest in this ambitious direction, I am willing to lean towards a Weak Accept to encourage further research. However, I still have significant concerns about the framework's current practical deployment and privacy guarantees, which I hope the Area Chair will take into careful consideration.

---

> > > ### Author Response · Authors · 2026-04-02
> > >
> > > Thank you for your constructive feedback and for recognizing the significance of our problem setting. We really appreciate your willingness to lean towards a Weak Accept. We would like to emphasize that the core contribution of this work lies in addressing the critical challenge of unlearning within continual learning scenarios. We believe this framework can be further enhanced by integrating lightweight model architectures in our future research. Thank you very much again and have a nice day！

---

### Official Review · Reviewer_q5n6 · 2026-03-12

**Soundness:** 2
**Presentation:** 3
**Significance:** 2
**Originality:** 2
**Overall Recommendation:** 4
**Confidence:** 2

**Summary:**

This paper proposes DCU, a decentralized continual learning framework that supports on-demand class unlearning without storing historical data. The method extracts class prototypes using diffusion models to guide continual learning and enables selective class removal through prototype-guided synthetic data and parameter adjustment. Experiments on decentralized benchmarks demonstrate that DCU can retain knowledge of remaining classes while effectively forgetting target classes.

**Compliance With Llm Reviewing Policy:**

Affirmed.

**Final Justification:**

Thanks for the clarification by authors. I raised my score.

**Key Questions For Authors:**

Please refer to questions in Weaknesses

**Limitations:**

Please refer to Weaknesses

**Strengths And Weaknesses:**

**Strengths**

1. The paper addresses decentralized continual learning with on-demand class unlearning.

2. The proposed framework integrates continual learning and unlearning in a unified decentralized setting, which is practically relevant for privacy-sensitive scenarios.

3. The use of diffusion-generated class prototypes provides a data-free mechanism for knowledge reconstruction and removal.

4. The paper evaluates both knowledge retention and forgetting performance, which is appropriate for unlearning settings.

**Weaknesses**

1. The technical novelty is limited with combining existing ideas from prototype learning, synthetic data generation, and unlearning.

2. The experimental evaluation is relatively limited, relying on small-scale datasets (MNIST variants and CIFAR-100 subsets). How is the performance of this model in ImageNet family datasets?

3. The paper lacks comprehensive analysis of communication cost and scalability, which are critical in decentralized learning settings.

4. The practical cost of diffusion-based prototype generation is not comprehensively discussed or evaluated.

---

> ### Author Rebuttal · Authors · 2026-03-31
>
> Dear reviewer, we sincerely appreciate your valuable comments and constructive suggestions.
>
> **Response to W1:** We maintain that the core contribution lies not merely in the combination of techniques, but in the novel unified paradigm we establish. Specifically, we demonstrate that by utilizing disposable, synthetic prototypes derived from frozen diffusion models, our framework can simultaneously address two fundamentally opposing objectives: (1) retaining historical knowledge for continual learning via distillation, and (2) facilitating precise erasure of specific knowledge for unlearning via boundary adjustment. This dual capability, achieved without storing raw data, represents the key conceptual advancement of our work.
>
> **Response to W2:** We additionally conduct partial experiments on the first 30 classes of ImageNet, following the same setting used for the RGB dataset.
>
> | | $cl_{0:9}$|$cl_{10:19}$|$ul_{2}$|$cl_{20:29}$|$ul_{20}$|
> |--|--|--|--|--|--|
> |lwf_au|44.4| 32.4|--| 23.87| 20.62|
> |target_au|44.4| 36.0| 35.47| 25.93| 26.29|
> |clmul| 44.4| 31.1| 21.79| 27.86| 24.43|
> |Ours | 51.24| 37.88| 36.46| 27.86| 27.47|
>
> **Response to W3&W4:**  In terms of communication cost, each class prototype is a lightweight 1280-dimensional vector, occupying only 0.005 MB. Since the embedding size is fixed, the storage and communication costs are independent of the number of clients. Regarding time cost, we primarily report training-related per-client timings on the RGB dataset. Specifically, the prototype training time, model training time, and unlearning time, as summarized in the table. The data generation for each class may incur considerable overhead due to our use of a naive DDIM sampling strategy. However, this can be mitigated through inference acceleration techniques.
>
> ||prototype training|learning|unlearning|data generation for each class
> |--|--|--|--|--|
> |time cost(min)|$\approx$ 5| $\approx$ 14.5|$\approx$ 0.15|$\approx$ 3.6
> |rounds|10|50|-|-|
>
> Thank you again for the valuable comments. We hope you might find the response satisfactory.

---

> > ### Author Rebuttal · Reviewer_q5n6 · 2026-04-05
> >
> > Thanks for the clarification. I will raise my score.

---

> > > ### Author Response · Authors · 2026-04-05
> > >
> > > Thank you for the constructive feedback and for raising your score. We appreciate your time and the recognition of our efforts to improve the manuscript.

---

### Decision · Program_Chairs · 2026-04-30

**Decision:**

Accept (regular)

**Comment:**

The paper focuses on decentralized continual learning and on-demand unlearning. The problem setting is novel and practical. The paper proposes to leverage a diffusion model to represent a class, and use the data generated by the diffusion model to support decentralized continual learning and unlearning.

The paper has some limitations in its novelty and scalability of experiments. Moreover, reviewers have pointed out that a shared pretrained diffusion model is still needed, which may partially weaken the claim of decentralization or at least incur additional communication cost. However, the paper has provided a novel problem setting which may motivate further exploration in this direction. The rebuttal has provided supplementary empirical studies to make the work more complete.